# Scalable DBSCAN with Random Projections

**HaoChuan Xu**
School of Computer Science
University of Auckland
hxu612@aucklanduni.ac.nz

**Ninh Pham**
School of Computer Science
University of Auckland
ninh.pham@auckland.ac.nz

## Abstract

We present *sDBSCAN*, a scalable density-based clustering algorithm in high dimensions with cosine distance. sDBSCAN leverages recent advancements in random projections given a significantly large number of random vectors to quickly identify core points and their neighborhoods, the primary hurdle of density-based clustering. Theoretically, sDBSCAN preserves the DBSCAN's clustering structure under mild conditions with high probability. To facilitate sDBSCAN, we present *sOPTICS*, a scalable visual tool to guide the parameter setting of sDBSCAN. We also extend sDBSCAN and sOPTICS to L2, L1, $\chi^2$, and Jensen-Shannon distances via random kernel features. Empirically, sDBSCAN is significantly faster and provides higher accuracy than competitive DBSCAN variants on real-world million-point data sets. On these data sets, sDBSCAN and sOPTICS run in a few minutes, while the scikit-learn counterparts and other clustering competitors demand several hours or cannot run on our hardware due to memory constraints. Our code is available at https://github.com/NinhPham/sDbscan.

## 1 Introduction

DBSCAN [1] is one of the most fundamental clustering algorithms with many applications in data mining and machine learning [2]. It has been featured in several data analysis tool kits, including scikit-learn in Python, ELKI in Java, and CRAN in R. In principle, DBSCAN connects neighboring points from nearby high-density areas to form a cluster where the high density is decided by a sufficiently large number of points in the neighborhood. DBSCAN is parameterized by $(\varepsilon, minPts)$ where $\varepsilon$ is the distance threshold to govern the point's neighborhood and to connect nearby areas; and $minPts$ is the density threshold to identify high-density areas.

Apart from other popular clustering algorithms, including $k$-means variants [3, 4] and spectral clustering [5], DBSCAN is non-parametric. It can find the number of clusters, detect arbitrary clustering shapes and sizes, and work on any arbitrary distance measure.

Given a distance measure, DBSCAN has two primary steps, including (1) finding the $\varepsilon$-neighborhood (i.e. points within a radius $\varepsilon$) for every point to discover the density areas surrounding the point and (2) forming the cluster by connecting neighboring points. The first step is the main bottleneck as finding $\varepsilon$-neighborhoods for all points requires a worst-case $O(n^2)$ time for a data set of $n$ points in high-dimensions [6, 7]. This limits the applications of DBSCAN on modern million-point data sets.

Another hurdle of $(\varepsilon, minPts)$-DBSCAN is the choice of $\varepsilon$, which highly depends on the data distribution and distance measure. While $minPts$ is easier to set for smoothing the density estimate, DBSCAN's outputs are susceptible to $\varepsilon$, especially in high dimensions where the range of $\varepsilon$ is very sensitive. For instance, when applying DBSCAN with cosine distance on the Pamap2 data set, changing $\varepsilon$ by just 0.005 can diminish the clustering accuracy by 10%. In practice, ones often need to compute an $\varepsilon$-neighborhood for each point given a large value of $\varepsilon$, and use them to explore the quality of clustering structures over smaller values of $\varepsilon$. Using large $\varepsilon$ causes $O(n^2)$ memory bottleneck

38th Conference on Neural Information Processing Systems (NeurIPS 2024).

as the $\varepsilon$-neighborhood of one point might need $O(n)$ space. We observe that on our hardware, such memory constraint is the primary hurdle limiting the current scikit-learn implementation on million-point data sets. Therefore, it is essential to not only develop scalable versions of DBSCAN, but also to provide feasible tools to guide its parameter setting on large-scale data sets.

**Prior arts on scaling up DBSCAN.** Due to the quadratic time bottleneck of DBSCAN in high-dimensional space, researchers study efficient solutions to scale up the process of identifying the neighborhood for each point in exact and approximate manners.

Exact DBSCAN approaches [8, 9, 10] partition the data set into several subsets and iteratively extract and refine clusters from these subsets by performing additional $\varepsilon$-neighborhood queries on a small set of important points. Other grid-based methods on L2 [6, 11] efficiently identify neighborhood areas by confining the neighbor search exclusively to neighboring grids. However, these approaches still have worst-case quadratic time or their complexity grows exponentially to the dimension. Random projections [12, 13] have been used to build grid-based or tree-based indexes for faster approximate $\varepsilon$-neighborhoods on L2, though they do not offer theoretical guarantees on the clustering accuracy.

Instead of finding the $\varepsilon$-neighborhood for every point, DBSCAN++ [14] finds $\varepsilon$-neighborhoods for a subset of random points chosen through uniform sampling or $k$-centering methods. sngDBSCAN [15] approximates the $\varepsilon$-neighborhood for every point by computing the distance between the point to a subset of random points. Though sampling-based DBSCAN variants are simple and efficient, they offer statistical guarantees on the clustering accuracy via level set estimation [16] that requires many strong assumptions on the data distribution. Moreover, selecting suitable parameter values (especially for $\varepsilon$) for sampling-based approaches is challenging due to the nature of sampling.

**DBSCAN's parameter setting guideline with OPTICS.** OPTICS [17] attempts to mitigate the problem of selecting relevant $\varepsilon$ by linearly ordering the data points such that close points become neighbors in the ordering. Besides the cluster ordering, OPTICS also computes a reachability distance for each point. The dendrogram provided by OPTICS visualizes the density-based clustering results corresponding to a broad range of $\varepsilon$, which indicates a relevant range of $\varepsilon$ for DBSCAN.

Like DBSCAN, OPTICS requires the $\varepsilon$-neighborhood for every point, leading to an $O(n^2)$ time complexity. In practice, OPTICS often needs a large $\varepsilon$ to discover clustering structures on a wide range of $\varepsilon$. Such large $\varepsilon$ demands $O(n^2)$ memory as the $\varepsilon$-neighborhood of one point needs $O(n)$ space. Such memory constraint is infeasible for million-point data sets.

**Contribution.** Inspired by sampling approaches, we observe that the exact $\varepsilon$-neighborhood for every point is unnecessary to form and visualize the density-based clustering results. Our approach, named *sDBSCAN*, first builds a *lightweight* random projection-based index with a sufficiently large number of projection vectors. Utilizing the asymptotic property of the extreme order statistics associated with some specific random vectors [18], sDBSCAN can select high-quality candidates to identify $\varepsilon$-neighborhoods with *theoretical* guarantees. sDBSCAN provably outputs DBSCAN's clustering structure similar to DBSCAN on cosine distance under mild conditions from data distribution. These conditions are much weaker than the ones used on recent sampling-based DBSCAN [14, 15]. Empirically, sDBSCAN runs significantly faster than several competitive DBSCAN variants while achieving similar DBSCAN clustering accuracy on million-point data sets.

To further facilitate sDBSCAN, we propose *sOPTICS*, a scalable OPTICS derived from the random projection-based indexing, to guide the parameter setting. We also extend sDBSCAN and sOPTICS to other popular distance measures, including L2, L1, $\chi^2$, and Jensen-Shannon (JS), that allow random kernel features [19, 20].

**Scalability.** Both sDBSCAN and sOPTICS are scalable and multi-thread friendly. Multi-threading sDBSCAN and sOPTICS take *a few minutes* to cluster and visualize clustering structures for million-point data sets while the scikit-learn counterparts cannot run on our hardware. On Mnist8m with 8.1 million points, sDBSCAN gives 38% accuracy (measuring by the normalized mutual information NMI), running in 15 minutes on a *single* machine of 2.2GHz 32-core (64 threads) AMD processor with 128GB of DRAM. In contrast, kernel $k$-means [21] achieves 41% accuracy with Spark, running in 15 minutes on a *supercomputer* with 32 nodes, each of which has two 2.3GHz 16-core (32 threads) Haswell processors and 128GB of DRAM.

## 2 Preliminary

We briefly describe DBSCAN [1], OPTICS [17], and the connection with approximate near neighbor search (ANNS). We present a recent advanced random projection method [22, 23] for ANNS on the extreme setting where the number of random projection vectors is sufficiently large. The data structures inspired by these approaches can scale up DBSCAN and OPTICS on large-scale data sets.

### 2.1 DBSCAN

DBSCAN is a density-based approach that links nearby dense areas to form the cluster. For a distance measure $dist(\cdot, \cdot)$, DBSCAN has two parameters $\varepsilon$ and $minPts$. Given the data set $\mathbf{X}$, for each point $\mathbf{q} \in \mathbf{X}$, DBSCAN executes a *range reporting query* $B_\varepsilon(\mathbf{q})$ that finds *all* points $\mathbf{x} \in \mathbf{X}$ within the $\varepsilon$-neighborhood of $\mathbf{q}$, i.e. $B_\varepsilon(\mathbf{q}) = \{\mathbf{x} \in \mathbf{X} \,|\, dist(\mathbf{x}, \mathbf{q}) \leq \varepsilon\}$. Based on the size of the range query result, DBSCAN determines $\mathbf{q}$ as *core* if $|B_\varepsilon(\mathbf{q})| \geq minPts$; otherwise, *non-core* points. We will call the $\varepsilon$-neighborhood $B_\varepsilon(\mathbf{q})$ as the *neighborhood* of $\mathbf{q}$ for short.

DBSCAN forms clusters by connecting core points and their neighborhoods where two core points $\mathbf{q}_1$ and $\mathbf{q}_2$ are connected if $\mathbf{q}_1 \in B_\varepsilon(\mathbf{q}_2)$. A non-core point belonging to a core point's neighborhood will be considered a *border* point and share the core point's label. Non-core points not belonging to any core point's neighborhood will be classified as *noise*. Indeed, DBSCAN forms a graph $G$ that connects $n$ points together [7, 14]. $G$ will have several disconnected components corresponding to the cluster structure. Each connected component contains connected core points and their neighborhoods, as shown in Alg. 1.

---
**Algorithm 1** DBSCAN

1: **Inputs**: $\mathbf{X}, \varepsilon, minPts$, the set $C = \{(\mathbf{q}, B_\varepsilon(\mathbf{q})) \,|\, \mathbf{q}$ is core$\}$
2: $G \leftarrow$ initialize empty graph
3: **for** each $\mathbf{q} \in C$ **do**
4:     Add an edge (and possibly vertices) in $G$ from $\mathbf{q}$ to all *core* points in $B_\varepsilon(\mathbf{q})$
5:     Add an edge (and possibly vertices) in $G$ from $\mathbf{q}$ to *non-core* points $\mathbf{x} \in B_\varepsilon(\mathbf{q})$ if $\mathbf{x}$ is not connected
6: **return** connected components of $G$

---

### 2.2 OPTICS

OPTICS [17] attempts to mitigate the problem of selecting relevant $\varepsilon$ by linearly ordering the data points such that close points become neighbors in the ordering. For each point $\mathbf{q} \in \mathbf{X}$, OPTICS computes a reachability distance from its closest core point. The cluster ordering and reachability distance of each point are used to construct a reachability-plot dendrogram that visualizes the density-based clustering results corresponding to a broad range of $\varepsilon$. Valleys in the reachability-plot are considered as clustering indicators. The OPTICS's algorithm is detailed in the appendix.

Given a pair $(\varepsilon, minPts)$, OPTICS first identifies the core points, the neighborhoods of core points, and then computes the core distance of every point, defined as below.

$$coreDist(\mathbf{q}) = \begin{cases} \infty & \text{if } \mathbf{q} \text{ is non-core,} \\ minPts - \text{NN distance} & \text{otherwise.} \end{cases}$$

Then, OPTICS iterates $\mathbf{X}$, and for each $\mathbf{x} \in \mathbf{X}$, computes the *smallest* reachability distance, defined by $\mathbf{x}.reach$, between $\mathbf{x}$ and the *processed* core points so far. The point with the minimum reachability distance will be processed first and inserted into the cluster ordering $O$. The reachability distance $reachDist(\mathbf{x}, \mathbf{q})$ is defined as follows.

$$reachDist(\mathbf{x}, \mathbf{q}) = \begin{cases} \infty & \text{if } \mathbf{q} \text{ is non-core,} \\ max(coreDist(\mathbf{q}), dist(\mathbf{x}, \mathbf{q})) & \text{otherwise.} \end{cases}$$

For a core point $\mathbf{q}$, $reachDist(\mathbf{x}, \mathbf{q})$ is $dist(\mathbf{x}, \mathbf{q})$ if $\mathbf{x}$ is not belonging to $minPts-$NN of $\mathbf{q}$. Among several core points whose neighborhood contains $\mathbf{x}$, OPTICS tends to seek the *smallest* reachability distance, i.e. $\mathbf{x}.reach$, for $\mathbf{x}$ from these core points, which reflects the distance between $\mathbf{x}$ and its nearby cluster. Therefore, points tend to be grouped with its neighborhood to form a cluster. A sharp decrease of $\mathbf{x}.reach$ of the point $\mathbf{x}$ in the group indicates that we are processing points in denser regions, and a slight increase indicates that we are processing points in sparser regions. This creates valleys on the dendrograms reflecting the number of clusters where points downwards the valley floor are on denser regions while points upwards the valley head are on sparser regions. By selecting $\varepsilon$ to separate valleys provided by OPTICS, DBSCAN can achieve the peak of accuracy.

**Time and space complexity.** Similar to DBSCAN, the running time of OPTICS is $O(n^2)$ makes it impractical on large-scale data sets. Fast implementations of OPTICS with large values of $\varepsilon$ might require $O(n^2)$ memory to store the matrix distance between the core points and their neighborhood points. Such implementations are infeasible for large $n$.

### 2.3 Random projection-based neighborhood preservation

Since the primary bottleneck of DBSCAN is to find core points and their neighborhoods by executing $n$ range queries, reducing the computational cost of this step will significantly improve the performance. Scaling up DBSCAN poses the need for *lightweight* indexing data structures, ideally with linear construction time, to approximately answer $n$ range queries. Heavyweight graph-based indexes [24, 25] with $O(n^2)$ construction time and locality-sensitive hashing (LSH)-based indexes [23, 26] with subquadratic construction time will dominate the clustering time.

We elaborate on the recent work, called CEOs [22], that studies extreme order statistics properties of random projection methods to approximate inner product. Given $D$ random vectors $\mathbf{r}_i \in \mathbb{R}^d$, $i \in [D]$, whose coordinates are randomly selected from the standard normal distribution $N(0, 1)$, and the sign function $\text{sgn}(\cdot)$. CEOs randomly projects $\mathbf{x}$ and $\mathbf{q}$ onto these $D$ Gaussian random vectors. It studies the behavior of the projection values on specific random vectors that are closest or furthest to $\mathbf{q}$, e.g., $\arg\max_{\mathbf{r}_i} |\mathbf{q}^\top \mathbf{r}_i|$. Given a sufficiently large $D$, the projection values on the closest or furthest random vector to $\mathbf{q}$ approximately preserve $\mathbf{x}^\top \mathbf{q}$. The proof is described in the appendix.

**Lemma 1.** *[22] For two points* $\mathbf{x}, \mathbf{q} \in \mathcal{S}^{d-1}$ *and significantly large $D$ random vectors* $\mathbf{r}_i$*, w.l.o.g. we let* $\mathbf{r}_* = \arg\max_{\mathbf{r}_i} |\mathbf{q}^\top \mathbf{r}_i|$*. Then, we have*

$$\mathbf{x}^\top \mathbf{r}_* \xrightarrow{D} N\left(\text{sgn}(\mathbf{q}^\top \mathbf{r}_*) \cdot \mathbf{x}^\top \mathbf{q} \sqrt{2\ln(D)}, 1 - (\mathbf{x}^\top \mathbf{q})^2\right). \tag{1}$$

As a geometric intuition, for significantly large $D$ random vectors, if $\mathbf{r}_*$ is closest or furthest to $\mathbf{q}$, the projection values of all points in $\mathbf{X}$ onto $\mathbf{r}_*$ tend to preserve their inner product order with $\mathbf{q}$. For a constant $k > 0$, Lemma 1 also holds for the top-$k$ closest/furthest vectors to $\mathbf{q}$ due to the asymptotic property of extreme normal order statistics [18, 22]. Therefore, by maintaining a few points that are closest/furthest to random vectors, we can approximate the neighborhood of each point accurately and efficiently. We will utilize this observation to significantly reduce the cost of identifying core points and, hence, the running time of DBSCAN and OPTICS in high dimensions.

## 3 Scalable density-based clustering with random projections

We first present simple and scalable DBSCAN, called *sDBSCAN*, with cosine distance. We then leverage well-known random feature embeddings [19, 20] to extend our proposed *sDBSCAN* to other popular distance measures, including L1 and L2 metrics, and widely used similarity measures for image data, including $\chi^2$ and Jensen-Shannon (JS). We also present *sOPTICS* to select relevant values of $\varepsilon$ for sDBSCAN. The detailed discussion and complexity of sOPTICS are in the appendix.

### 3.1 sDBSCAN: A simple and scalable density-based clustering

For each point $\mathbf{q} \in \mathbf{X} \subset \mathcal{S}^{d-1}$, we compute $D$ random projection values $\mathbf{q}^\top \mathbf{r}_i$ where $i \in [D]$. W.l.o.g., we let $\mathbf{r}_* = \arg\max_{\mathbf{r}_i} |\mathbf{q}^\top \mathbf{r}_i| = \arg\max_{\mathbf{r}_i} \mathbf{q}^\top \mathbf{r}_i$. Lemma 1 indicates that, given a sufficiently large $D$, points closest to $\mathbf{r}_*$ tend to have smaller distances to $\mathbf{q}$, and hence are important to discover $\mathbf{q}$'s neighborhood. Hence, computing the distance between $\mathbf{q}$ to the top-$m$ points closest to $\mathbf{r}_*$ where $m = O(minPts)$ suffices to ensure whether or not $\mathbf{q}$ is a core point.

Algorithm 2 shows how we preprocess the data set. For each point $\mathbf{q} \in \mathbf{X}$, we keep top-$k$ closest and furthest random vectors. For each random vector $\mathbf{r}_i$, we keep top-$m$ closest and furthest points. Algorithm 3 identifies a core point $\mathbf{q}$ and its approximate neighborhood by computing the distance between $\mathbf{q}$ and $2km$ points associated to the $k$ closest and $k$ furthest random vectors to $\mathbf{q}$. sDBSCAN and sOPTICS are essential DBSCAN and OPTICS using the outputs of Algorithm 3 as inputs.

**Scalability.** Given $m = O(minPts)$, sDBSCAN and sOPTICS need $O(k \cdot minPts)$ distance computations, compared to $O(n)$ of the exact solution, to identify a core point $\mathbf{q}$ and its neighborhood

---

**Algorithm 2** Preprocessing

---

1: **Inputs**: $\mathbf{X} \subset \mathcal{S}^{d-1}$, $D$ random vectors $\mathbf{r}_i$, $k, m = O(minPts)$
2: **for** each $\mathbf{q} \in \mathbf{X}$, compute and store top-$k$ closest and top-$k$ furthest vectors $\mathbf{r}_i$ to $\mathbf{q}$.
3: **for** each random vector $\mathbf{r}_i$, compute and store top-$m$ closest and top-$m$ furthest points to $\mathbf{r}_i$.

---

---

**Algorithm 3** Finding core points and their approximate neighborhoods

---

1: **Inputs**: $\mathbf{X} \subset \mathcal{S}^{d-1}$, $D$ random vectors $\mathbf{r}_i$, $k, \varepsilon, m = O(minPts)$
2: Initialize an empty set $\widetilde{B}_\varepsilon(\mathbf{q})$ for each $\mathbf{q} \in \mathbf{X}$
3: **for** each $\mathbf{q} \in \mathbf{X}$ **do**
4:    **for** each $\mathbf{r}_i$ from top-$k$ closest (or furthest) random vectors of $\mathbf{q}$ **do**
5:       **for** each $\mathbf{x}$ from top-$m$ closest (or furthest) points of $\mathbf{r}_i$ **do**
6:          **if** $dist(\mathbf{x}, \mathbf{q}) \leq \varepsilon$ **then**
7:             Insert $\mathbf{x}$ into $\widetilde{B}_\varepsilon(\mathbf{q})$ and insert $\mathbf{q}$ into $\widetilde{B}_\varepsilon(\mathbf{x})$
8: **for** each $\mathbf{q} \in \mathbf{X}$ **do**
9:    **if** $|\widetilde{B}_\varepsilon(\mathbf{q})| \geq minPts$ **then**
10:       Output $\mathbf{q}$ as a core point and $\widetilde{B}_\varepsilon(\mathbf{q})$ as an approximate $B_\varepsilon(\mathbf{q})$ for DBSCAN (Alg. 1)
11:       Output $dist(\mathbf{x}, \mathbf{q})$ for each $\mathbf{x} \in \widetilde{B}_\varepsilon(\mathbf{q})$ for OPTICS (Alg. 6)

---

---

**Algorithm 4** sDBSCAN

---

1: **Inputs**: $\mathbf{X} \subset \mathcal{S}^{d-1}$, $D$ random vectors $\mathbf{r}_i$, $\varepsilon, minPts$
2: Call Algorithm 2 for preprocessing with $m = O(minPts)$
3: Call Algorithm 3 to find the set $C = \{(\mathbf{q}, \widetilde{B}_\varepsilon(\mathbf{q})) \,|\, \mathbf{q} \text{ is identified as core}\}$
4: Call DBSCAN given the output $C$ from Algorithm 3

---

---

**Algorithm 5** sOPTICS

---

1: **Inputs**: $\mathbf{X} \subset \mathcal{S}^{d-1}$, $D$ random vectors $\mathbf{r}_i$, $\varepsilon, minPts$
2: Call Algorithm 2 for preprocessing with $m = O(minPts)$
3: Call Algorithm 3 to find the set $C = \{(\mathbf{q}, \widetilde{B}_\varepsilon(\mathbf{q}),$
   $\{dist(\mathbf{x}, \mathbf{q}) \text{ for each } \mathbf{x} \in \widetilde{B}_\varepsilon(\mathbf{q})\}) \,|\, \mathbf{q} \text{ is identified as core}\}$
4: Use the $minPts-$NN distance between $\mathbf{q}$ and $\mathbf{x} \in \widetilde{B}_\varepsilon(\mathbf{q})$ as $coreDist(\mathbf{q})$ for each identified core point $\mathbf{q}$
5: Call OPTICS (Alg. 6) given the output $C$ from Algorithm 3

---

subset $\widetilde{B}_\varepsilon(\mathbf{q}) \subseteq B_\varepsilon(\mathbf{q})$. The memory usage to store $\widetilde{B}_\varepsilon(\mathbf{q})$ is also bounded by $O(k \cdot minPts)$. This makes sDBSCAN and sOPTICS scalable regarding both time and space complexity.

**Multi-threading.** Like DBSCAN, the main computational bottlenecks of sDBSCAN and sOPTICS are identifying core points and approximating their neighborhood. Fortunately, Algorithm 2 and 3 are fast and parallel-friendly. We only need to add `#pragma omp parallel` directive on the `for` loops to run in multi-threads. This enables sDBSCAN and sOPTICS to deal with million-point data sets in *minutes* while the scikit-learn counterparts take hours or cannot finish due to the memory constraints.

### 3.2 Theoretical analysis of sDBSCAN

In practice, setting $m = O(minPts)$ suffices to identify core points and approximate their neighborhoods to ensure the quality of sDBSCAN. However, to *theoretically* guarantee the quality of sDBSCAN, we need to adjust $m$ based on the data distribution since ensuring the density-based clustering quality without any information about the data distribution is hard [14, 15].

As in the unit sphere, if $L_2(\mathbf{x}, \mathbf{q}) \leq \varepsilon$ then $\mathbf{x}^\top \mathbf{q} \geq 1 - \varepsilon^2/2$, we make a change in our algorithm. For each random vector $\mathbf{r}_i$, we maintain two sets $S_i = \{\mathbf{x} \in \mathbf{X} \,|\, \mathbf{x}^\top \mathbf{r}_i \geq (1 - \varepsilon^2/2)\sqrt{2\ln(D)}\}$ and $R_i = \{\mathbf{x} \in \mathbf{X} \,|\, \mathbf{x}^\top \mathbf{r}_i \leq -(1 - \varepsilon^2/2)\sqrt{2\ln(D)}\}$, instead of keeping only top-$m$ closest/furthest points. In other words, for each random vector $\mathbf{r}_i$, the value of $m$ is set *adaptively* to the data

distributed around $\mathbf{r}_i$. For a point $\mathbf{q}$ and its closest random vector $\mathbf{r}_*$, if we compute $L_2(\mathbf{x}, \mathbf{q})$ for all $\mathbf{x} \in S_*$ where $S_* = \{\mathbf{x} \in \mathbf{X} \mid \mathbf{x}^\top \mathbf{r}_* \geq (1 - \varepsilon^2/2)\sqrt{2 \ln(D)}\}$, then any $\mathbf{x} \in B_\varepsilon(\mathbf{q})$ is found with a probability at least $1/2$ due to the Gaussian distribution in Eq. (1). By computing the distance between $\mathbf{q}$ and all points in $S_i$ or $R_i$ corresponding to the top-$k$ closest/furthest random vectors $\mathbf{r}_i$ of $\mathbf{q}$, we can boost this probability to $s = 1 - (1/2)^{2k}$ due to the asymptotic independence among these $2k$ vectors [18, 22].

By connecting core points and their associated border points, DBSCAN outputs an $\varepsilon$-neighborhood graph with $l$ disconnected components $G_1, G_2, \ldots, G_l$ corresponding to $l$ density-based clusters $C_1, C_2, \ldots, C_l$. Hence, by maintaining $S_i$ and $R_i$ for each vector $\mathbf{r}_i$, sDBSCAN forms a subgraph $G_i' \subseteq G_i$, $i = 1, \ldots, l$, by sampling each edge of $G_i$ with probability at least $s$. We now use the following lemma [27] to guarantee the connectivity of subgraphs $G_i'$ provided by sDBSCAN.

**Lemma 2.** *Let $G$ be a graph of $n$ vertices with min-cut $t$ and $0 < \delta < 1$. There exists a universal constant $c$ such that if $s \geq \frac{c(\log(1/\delta) + \log(n))}{t}$, then with probability at least $1 - \delta$, the graph $G'$ derived by sampling each edge of $G$ with probability $s$ is connected.*

Since DBSCAN connects core points, there exists a constant $t > 0$ such that, for any pair of nearby core points $\mathbf{q}_1, \mathbf{q}_2$, if $L_2(\mathbf{q}_1, \mathbf{q}_2) \leq \varepsilon$, then $B_\varepsilon(\mathbf{q}_1)$ and $B_\varepsilon(\mathbf{q}_2)$ share *at least* $t$ common core points. We assume that $t$ will not be small to ensure that the density-based clusters will not become arbitrarily thin anywhere. Given that DBSCAN produces $l$ disconnected graphs $G_1, G_2, \ldots, G_l$, it is clear that $t$ will be the lower bound of the min-cut of any $G_i$. By applying Lemma 2 on each $G_i$, we state our main result.

**Theorem 1.** *Let $G_1, G_2, \ldots, G_l$ be connected subgraphs produced by DBSCAN where each $G_i$ corresponds to a cluster $C_i$ with $n_i$ core points. Assume that any pair of nearby core points $\mathbf{q}_1, \mathbf{q}_2$, if $L_2(\mathbf{q}_1, \mathbf{q}_2) \leq \varepsilon$, then $B_\varepsilon(\mathbf{q}_1)$ and $B_\varepsilon(\mathbf{q}_2)$ share at least $t$ common core points. There exists a constant $c$ such that if $t\left(1 - (1/2)^{2k}\right) \geq c\left(\log(1/\delta) + \log(n_i)\right)$ for $i \in [l]$, sDBSCAN will recover $G_1, G_2, \ldots, G_l$ as clusters with probability at least $1 - \delta$.*

Theorem 1 indicates that when the cluster $C_i$ of $n_i$ core points is not thin anywhere, i.e. the common neighborhood of any two nearby core points has at least $t = O(\log(n_i))$ core points, sDBSCAN can recover $C_i$ with probability $1 - 1/n_i$. While our statistical guarantee is inspired by [15], we do not need any strong assumption about the data distribution as used in [15]. However, it comes with the cost of maintaining larger neighborhoods around the random vector $\mathbf{r}_i$ (i.e. $S_i$, $R_i$) that causes significant computational resources.

In practice, a core point $\mathbf{q}$ is often surrounded by many other core points in a dense cluster. Therefore, instead of maintaining the sets $S_i$, $R_i$ for each random vector $\mathbf{r}_i$, we can keep the top-$m$ points of these sets where $m = O(minPts)$. This practical setting substantially reduces the memory usage and running time of sDBSCAN without degrading clustering results. The next subsection justifies it.

### 3.3 From theory to practice

We observe that for any core point $\mathbf{q}$ regarding $(\varepsilon, minPts)$, $|B_\varepsilon(\mathbf{q})|$ is often substantially larger than $minPts$. We will present heuristics to improve the performance of sDBSCAN.

**Identify core points with $m = O(minPts)$.** Given a core point $\mathbf{q}$, we denote by $\mathbf{x} \in B_\varepsilon(\mathbf{q})$ and $\mathbf{y} \in \mathbf{X} \setminus B_\varepsilon(\mathbf{q})$ any close and far away points to $\mathbf{q}$ regarding the distance threshold $\varepsilon$. W.l.o.g., we assume that $\mathbf{r}_* = \arg\max_{\mathbf{r}_i} \mathbf{q}^\top \mathbf{r}_i$, and let $X = \mathbf{x}^\top \mathbf{r}_*, Y = \mathbf{y}^\top \mathbf{r}_*$ be the projection values of $\mathbf{x}, \mathbf{y}$ on $\mathbf{r}_*$, respectively. From Eq. (1), $X - Y$ follows a Gaussian distribution whose variance is bounded by $\left(\sqrt{1 - (\mathbf{x}^\top \mathbf{q})^2} + \sqrt{1 - (\mathbf{y}^\top \mathbf{q})^2}\right)^2$. Let $\alpha_{\mathbf{xy}} = (\mathbf{x}^\top \mathbf{q} - \mathbf{y}^\top \mathbf{q}) / \left(\sqrt{1 - (\mathbf{x}^\top \mathbf{q})^2} + \sqrt{1 - (\mathbf{y}^\top \mathbf{q})^2}\right)$ and $\alpha_* = \arg\min_{\mathbf{x} \in B_\varepsilon(\mathbf{q}), \mathbf{y} \in \mathbf{X} \setminus B_\varepsilon(\mathbf{q})} \alpha_{\mathbf{xy}}$. As Lemma 1 holds for $k$ closest/furthest random vectors, assume that the event any point $\mathbf{x} \in B_\varepsilon(\mathbf{q})$ is ranked higher than all $\mathbf{y} \in \mathbf{X} \setminus B_\varepsilon(\mathbf{q})$ is independent, Lemma 3 justifies that $m = minPts$ suffices to identify $\mathbf{q}$ as a core point. The proof based on the tail bound of the Gaussian variable $X - Y$ and the union bound is left in the appendix.

**Lemma 3.** *Given $D = n^{1/k\alpha_*^2}$ for a given core point $\mathbf{q} \in \mathbf{X} \subset \mathcal{S}^{d-1}$ where $|B_\varepsilon(\mathbf{q})| \geq minPts$, maintaining top-$minPts$ points associated to $k$ closest/furthest vectors to $\mathbf{q}$ ensures $|\widetilde{B}_\varepsilon(\mathbf{q})| \geq minPts$, $\widetilde{B}_\varepsilon(\mathbf{q}) \subseteq B_\varepsilon(\mathbf{q})$ with probability at least $(1 - 1/n)^{minPts} \sim e^{-minPts/n}$.*

Lemma 3 holds given the dependence of $D$ on the data distribution around $\mathbf{q}$ (i.e. $\alpha_*$). We empirically observe that the distance gap between $\mathbf{q}$ and points inside and outside $B_\varepsilon(\mathbf{q})$ is significant, making $\alpha_*^2$ large. Hence, setting $D = 1,024, k = \{5, 10\}$ suffices to identify core points and enrich their neighborhoods, achieving high clustering quality on many real-world data sets.

**sDBSCAN-1NN to cluster misidentified border points.** As sDBSCAN focuses on identifying core points and approximating their neighborhoods, it misclassifies border points as noise. As the neighborhood size of detected core points is upper bound by $2km$, sDBSCAN might suffer accuracy loss on data sets with a significantly large number of border points. Denote by $C$ and $\overline{C}$ the set of clustered core points and noisy points found by sDBSCAN, we propose a simple heuristic to classify any $\mathbf{x} \in \overline{C}$. We will build a nearest neighbor classifier (1NN) where training points are *sampled* from $C$, and scale up this classifier with the CEOs-based estimation approach [22, Alg. 1]. We call sDBSCAN with the approximate 1NN heuristic as *sDBSCAN-1NN*.

In particular, for each sampled core point $\mathbf{q} \in C$, we recompute their random projection values with the stored $D$ random vector $\mathbf{r}_i$ as we do not keep these values after preprocessing. For each noisy point $\mathbf{x} \in \overline{C}$, we retrieve the *precomputed* top-$k$ closest/furthest vectors and use the projection values of $\mathbf{q}$ at these top-$k$ vectors to estimate $\mathbf{x}^\top \mathbf{q}$. To ensure this heuristic does not affect sDBSCAN time where $|C| \approx |\overline{C}| \approx n/2$, we sample $0.01n$ core points from $C$ to build the training set. Empirically, sDBSCAN time dominates this heuristic time, and the extra space usage to store $0.01nD$ projection values is negligible.

*Remark.* sDBSCAN quickly finds core points and assigns cluster labels for non-core points using a lightweight index. In contrast, LSH-based approaches [23, 28] need several LSH tables or multi-probes [29] that cause significant space or time overheads. Also, it seems non-trivial for multi-probe LSH to achieve DBSCAN's accuracy with guarantees as the collision probability of multi-probes is hard to control. Assigning label for identified non-core points with LSH seems non-trivial as these points might collide with many core points from different clusters.

**Extend to other distance measures.** We extend sDBSCAN to other popular distance measures, including L2, L1, $\chi^2$, and Jensen-Shannon (JS) via popular randomized kernel features [19, 20]. In particular, we study fast randomized feature mapping $f : \mathbb{R}^d \mapsto \mathbb{R}^{d'}$ such that $\mathbf{E}\left[f(\mathbf{x})^\top f(\mathbf{q})\right] = K(\mathbf{x}, \mathbf{q})$ where $K$ are Gaussian, Laplacian, $\chi^2$, and JS kernels. As we execute random projections on the constructed random features for each point and compute $dist(\mathbf{x}, \mathbf{y})$ using the original data, we only need a small extra space to store the randomness associated with $f$. These embeddings' extra costs are negligible compared to sDBSCAN. Detailed discussion is left in the appendix.

**Reduce the random projection costs.** We will use the Structured Spinners [26] that exploits Fast Hadamard Transform (FHT) to reduce the cost of Gaussian random projections. In particular, we generate 3 random diagonal matrices $\mathbf{D}_1, \mathbf{D}_2, \mathbf{D}_3$ whose values are randomly chosen in $\{+1, -1\}$. The random projection of $\mathbf{x}$ is simulated by the mapping $\mathbf{x} \mapsto \mathbf{H}\mathbf{D}_3\mathbf{H}\mathbf{D}_2\mathbf{H}\mathbf{D}_1\mathbf{x}$ where $\mathbf{H}$ is the Hadamard matrix. With FHT, the random projection can be simulated in $\mathcal{O}\left(D \log\left(D\right)\right)$ time and use additional $O(D)$ extra space to store random matrices.

### 3.4   The time and space complexity of sDBSCAN

We analyze the time and space complexity of sDBSCAN with $m = O(minPts)$.

**Time complexity.** With FHT, retrieving top-$k$ closest/furthest vectors for each $\mathbf{q} \in \mathbf{X}$, and top-$m$ closest/furthest points for each random vector $\mathbf{r}_i$ in the preprocessing (Alg. 2) runs in $O\left(nD \log\left(D\right) + nD \log\left(k\right) + nD \log\left(minPts\right)\right)$ time. Finding approximate neighborhoods for $n$ points (Alg. 3) runs in $O(dnk \cdot minPts)$ time since each point needs $O(k \cdot minPts)$ distance computations. For a constant $k$, sDBSCAN runs in $O(dn \cdot minPts + nD \log\left(D\right))$ time. When $D = o(n)$, sDBSCAN runs in *subquadratic* time.

Empirically, the cost of identifying core points and their neighborhood dominates the preprocessing cost due to the expensive random memory access of distance computations. Nevertheless, pre-

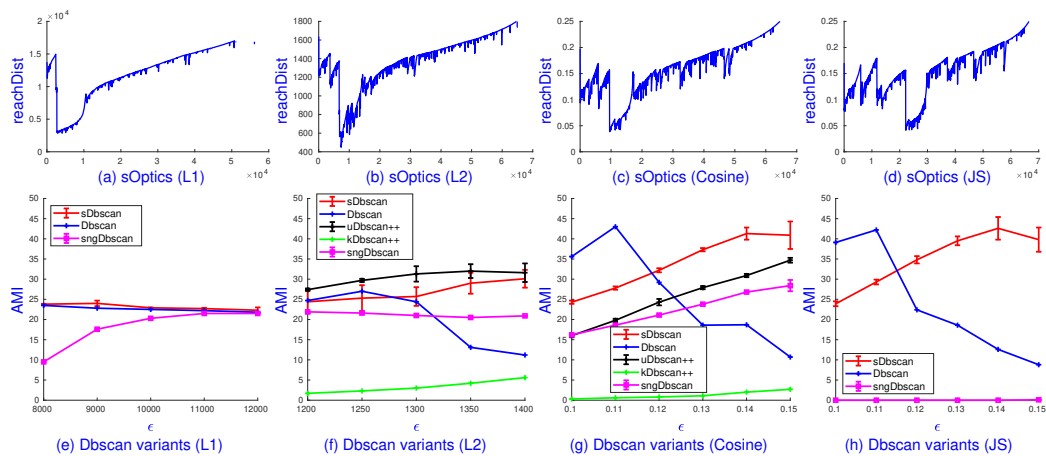

Figure 1: Top: sOPTICS's graphs on L1, L2, cosine, JS on Mnist. sOPTICS runs within **3 seconds** while scikit-learn OPTICS requires **1.5 hours** on L2. Bottom: AMI of DBSCAN variants on L1, L2, cosine, JS over the range of $\epsilon$ suggested by sOPTICS. Cosine and JS give the highest AMI.

processing and finding neighborhood steps are elementary to run in parallel. Our straightforward multi-threading implementation of sDBSCAN and sOPTICS shows a $10\times$ speedup with 64 threads.

**Space complexity.** sDBSCAN needs $O(nk + D \cdot minPts)$ extra space to store $O(k)$ closest/furthest vectors for each point, and $O(minPts)$ points closest/furthest to each random vector. When $D = o(n)$, sDBSCAN's additional memory is negligible compared to the data size $O(nd)$. This key feature makes sDBSCAN highly scalable on million-point data sets compared to standard DBSCAN and several kernel-based clustering [4, 5].

## 4   Experiment

We implement sDBSCAN and sOPTICS in C++ and compile with `g++ -O3 -std=c++17 -fopenmp -march=native`. We conducted experiments on Ubuntu 20.04.4 with an AMD Ryzen Threadripper 3970X 2.2GHz 32-core processor (64 threads) with 128GB of DRAM. We present empirical evaluations on the clustering quality compared to the ground truth (i.e. data labels) to verify our claims, including:

- sDBSCAN with the suggested parameter values provided by sOPTICS runs significantly faster and achieves competitive accuracy compared to other clustering algorithms.
- Multi-threading sDBSCAN and sOPTICS run in *minutes*, while the scikit-learn counterparts cannot run on million-point data sets due to memory constraints on our hardware.

Our competitors include pDBSCAN [6] as a representative grid-based approach, DBSCAN++ [14] and sngDBSCAN [15]. DBSCAN++ has two variants, including DBSCAN++ with uniform initialization (uDBSCAN++) that uses KD-Trees to speed up the search of core points and $k$-center initialization (kDBSCAN++). We also compare with multi-threading scikit-learn implementations of DBSCAN and OPTICS. To demonstrate the scalability and utility of sDBSCAN on other distance measures, we compare it with the result of kernel $k$-means in [21]. We found other clustering competitors could not work on million-point data sets given 128 GB of RAM, detailed in the appendix.

We use the adjusted mutual information (AMI) [30] to measure the clustering quality. Results on other measures [31], including normalized mutual information (NMI) and correlated coefficients (CC) are in the appendix. We conduct experiments on three popular data sets: Mnist ($n = 70,000, d = 784$, # clusters = 10), Pamap2 ($n = 1,770,131, d = 51$, # clusters = 18), and Mnist8m ($n = 8,100,000, d = 784$, # clusters = 10). All results are the average of 5 runs of the algorithms.

**Parameter settings.** We consider $minPts = 50$ for all experiments. sDBSCAN and sOPTICS use $D = 1024, m = minPts$. Randomized kernel embeddings use $\sigma = 2\varepsilon, d' = 1024$. We use $k = 5$ for Mnist and $k = 10$ for Pamap2 and Mnist8m. We first run sOPTICS to select a relevant range of

values of $\varepsilon$ for DBSCAN variants. For DBSCAN++ and sngDBSCAN variants, we set the sampling probability $p = 0.01$ for Mnist and Pamap2, and $p = 0.001$ for Mnist8m to have a similar number of distance computations with sDBSCAN. pDBSCAN uses $\rho = 0.001$. Experiments on the sensitivity of parameters $m, k, \sigma, d', minPts$ of sDBSCAN and sOPTICS are left in the appendix.

## 4.1 An ablation study of sOPTICS and sDBSCAN on Mnist

While DBSCAN on L2 is popular, the capacity to use arbitrary distances is an advantage of DBSCAN compared to other clustering algorithms. We use sOPTICS to visualize the cluster structure on several distance measures, including L1, L2, cosine, and JS on Mnist with $D = 1,024, m = minPts = 50, k = 5$. By using the average top-100 nearest neighbor distances of 100 random points to find the setting of $\varepsilon$ for sOPTICS, we set $\varepsilon = 1,800$ for L2, $\varepsilon = 18,000$ for L1, and $\varepsilon = 0.25$ for the others.

The top subfigures of Figure 1 show reachability-plot dendrograms of sOPTICS on 4 studied distance measures. Since points belonging to a cluster have a small reachability distance to their nearest neighbors, the number of valleys in the dendrograms reflects the cluster structure. Therefore, we can predict that cosine and JS provide higher clustering quality than L2 while L1 suffers low-quality clustering. Importantly, any $\varepsilon$ in the range $[0.1, 0.15]$ can differentiate the 4 valleys by cosine and JS while selecting a specific value to separate the 3 valleys by L2 is impossible. Note that 64-thread sOPTICS runs in less than *3 seconds* while 64-thread scikit-learn OPTICS demands *1.5 hours* on L2. Indeed, sOPTICS can output similar OPTICS graphs within 30 seconds, as shown in the appendix.

The bottom subfigures of Figure 1 show AMI scores of several DBSCAN variants over the recommended ranges of $\varepsilon$ by sOPTICS. Note that sDBSCAN and DBSCAN reach the peak on such ranges while sampling-based variants do not. sDbscan is superior on all 4 supported distances, except L2 by uDBSCAN++. While sDBSCAN reaches DBSCAN's accuracy of AMI 43% on cosine and JS, sngDBSCAN gives significantly lower accuracy on all 4 distances. uDBSCAN++ gives at most 32% AMI on L2 and cosine while kDBSCAN++ does not provide a meaningful result on the studied range values of $\varepsilon$. L2 and L1 distances show inferior performance on clustering compared to cosine and JS, as can be predicted from their corresponding sOPTICS graphs.

Table 1 summarizes the performance of studied DBSCAN variants on the *best $\varepsilon \in [0.1, 0.2]$* with cosine distance. On 1 thread, sDBSCAN runs nearly $2\times, 8\times$ and $10\times$ faster than sngDBSCAN, uDBSCAN++ and kDBSCAN++, respectively. 64-thread sDBSCAN runs nearly $10\times$ faster than 1-thread sDBSCAN and $100\times$ faster than 64-thread scikit-learn. Though pDBSCAN shares the same AMI with scikit-learn, its running time makes it infeasible for high-dimensional data sets.

## 4.2 Comparison on million-point data sets: Pamap2 and Mnist8m

This subsection compares the performance of sDBSCAN, DBSCAN++, and sngDBSCAN on million-point data sets. scikit-learn DBSCAN and pDBSCAN cannot finish after 4 hours. Our implemented sngDBSCAN runs faster than [15] and supports multi-threading. We use sOPTICS graphs in the appendix to select relevant ranges of $\varepsilon$. sOPTICS runs in **2 mins** and **11 mins** on Pamap2 and Mnist8m, respectively, significantly faster than DBSCAN++ and sngDBSCAN. The released DBSCAN++ does not support L1, $\chi^2$, JS and multi-threading while the rest are in multi-threading.

**Pamap2.** Figure 2 shows the performance of sDBSCAN compared to DBSCAN++ and sngDBSCAN. Given suggested ranges of $\varepsilon$ by sOPTICS, sDBSCAN's AMI peaks are consistently higher than that of sngDBSCAN but 5% lower than DBSCAN on L1 and L2. While sDBSCAN achieves the AMI peak on the studied ranges of $\varepsilon$ on 3 distances, the performance of the others is very different on different range of $\epsilon$ with cosine. We found that DBSCAN with $\varepsilon = 0.005$ returns AMI 47% but

Table 1: AMI on the best $\varepsilon \in [0.1, 0.2]$ and running time of **64-thread** scikit-learn vs. **1-thread** DBSCAN variants using cosine distance on Mnist. **64-thread** sDBSCAN runs in **0.9 seconds**.

| Alg. | scikit-learn | sDBSCAN | uDBSCAN++ | kDBSCAN++ | sngDBSCAN | pDBSCAN |
|------|------|------|------|------|------|------|
| AMI | **43%** | **43%** | **43%** | 7% | 33% | **43%** |
| Time | 86s | **8.8s** | 67s | 87s | 18s | 1.85 hours |
| $\varepsilon$ | 0.11 | 0.14 | 0.18 | 0.2 | 0.15 | 0.11 |

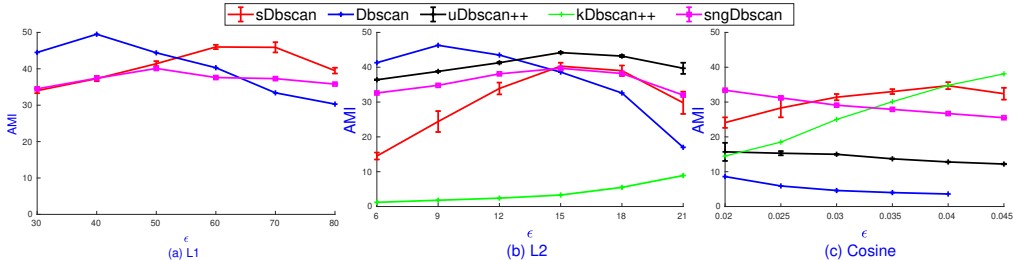

Figure 2: AMI comparison of DBSCAN variants on L1, L2 and cosine on Pamap2 over a wide range of $\varepsilon$ suggested by sOPTICS. sDBSCAN runs within **0.3 mins**, nearly $\mathbf{10\times, 10\times, 45\times, 100\times}$ faster than sngDBSCAN, uDBSCAN++, kDBSCAN++, and DBSCAN. L1 gives the highest AMI.

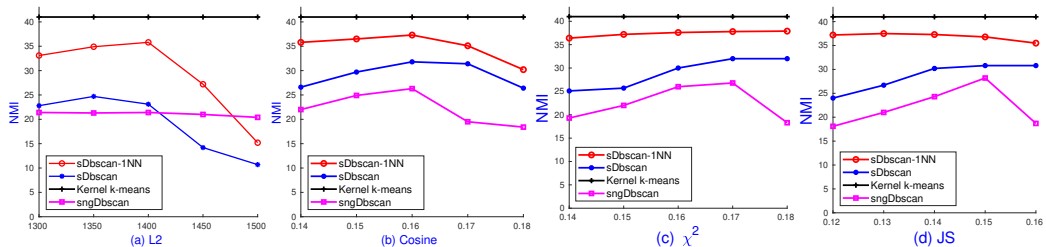

Figure 3: NMI comparison of DBSCAN variants on L2, cosine, $\chi^2$, and JS and kernel $k$-means on Mnist8m over a wide ranges of $\varepsilon$ suggested by sOPTICS. sDBSCAN and sDBSCAN-1NN runs within **10 mins** and **15 mins** while sngDBSCAN demands nearly **1 hour**. Kernel $k$-means ($k = 10$) [21] runs in **15 mins** on a supercomputer of 32 nodes, each has 64 threads and 128 GB of DRAM.

offers only 37% AMI on $\varepsilon = 0.01$. This explains the reliability of sOPTICS in guiding the parameter setting for sDBSCAN and the difficulty in selecting relevant $\varepsilon$ for other DBSCAN variants without any scalable visual tools.

**Mnist8m.** As DBSCAN++ with $p = 0.001$ could not finish after 4 hours, we only report sngDBSCAN. As we cannot run scikit-learn $k$-means++ or any kernel-based clustering [4, 5] with 128 GB of RAM, we use the result of 41% NMI given by a fast kernel $k$-means [21] running on a supercomputer with *32 nodes*, each of which has *two* 2.3GHz 16-core (32 threads) Haskell processors and 128GB of DRAM. This configuration of a *single* node is similar to our conducted machine. Figure 3 shows the NMI scores of sDBSCAN-1NN with 1-NN heuristic described in Subsection 3.3, sDBSCAN, sgnDBSCAN, and kernel $k$-means on Mnist8m. sDBSCAN-1NN shows superiority among DBSCAN variants. Its peak is at least 10% and 5% higher than sngDBSCAN and sDBSCAN, respectively, on studied measures. sDBSCAN consistently gives higher accuracy than sngDBSCAN with the most significant gap of 5% on $\chi^2$ and cosine. Note that sDBSCAN-1NN samples $0.01n$ core points, and uses CEOs to build the approximate 1NN classifier, the time overhead of this step is smaller than sDBSCAN's time. Indeed, sDBSCAN-1NN with $minPts = 100$ reaches 40% NMI on $\chi^2$, running within 15 minutes. Details of the running time comparison are in the appendix.

## 5 Conclusion

The paper presents a simple and scalable sDBSCAN for density-based clustering, and sOPTICS for interactive clustering exploration for high-dimensional data. By leveraging the neighborhood preservation of random projections, sDBSCAN preserves the DBSCAN's output with theoretical guarantees under mild conditions. We extend our proposed algorithms to other distance measures to facilitate density-based clustering on many applications with image data. Empirically, both sDBSCAN and sOPTICS are highly scalable, run in minutes on million-point data sets, and provide very competitive accuracy compared to other clustering algorithms. We hope sDBSCAN and sOPTICS will be featured on popular clustering libraries shortly.

## Acknowledgments and Disclosure of Funding

Ninh Pham is supported by Marsden Fund (MFP-UOA2226).

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

# A OPTICS and sOPTICS

## A.1 OPTICS

OPTICS [17] attempts to mitigate the problem of selecting relevant $\varepsilon$ by linearly ordering the data points such that close points become neighbors in the ordering. For each point $\mathbf{x} \in \mathbf{X}$, OPTICS computes a reachability distance from its closest core point. Each point's cluster ordering and reachability distance are used to construct a reachability-plot dendrogram that visualizes the density-based clustering results corresponding to a broad range of $\varepsilon$. Hence, given the setting of large $\varepsilon$, valleys in the reachability plot are considered as the clustering indicators.

In principle, given a pair $(\varepsilon, minPts)$, OPTICS first identifies the core points, their neighborhoods, and their core distances. Then, OPTICS iterates $\mathbf{X}$, and for each $\mathbf{x} \in \mathbf{X}$, computes the smallest reachability distance, called $\mathbf{x}.reach$, between $\mathbf{x}$ and the *processed* core points so far. The point with the minimum reachability distance will be processed first and inserted into the cluster ordering $O$.

The core distance, *coreDist*, and reachability distance, *reachDist*, are defined as follows.

$$coreDist(\mathbf{q}) = \begin{cases} \infty & \text{if } \mathbf{q} \text{ is non-core,} \\ minPts - \text{NN distance} & \text{otherwise.} \end{cases}$$

$$reachDist(\mathbf{x}, \mathbf{q}) = \begin{cases} \infty & \text{if } \mathbf{q} \text{ is non-core,} \\ max(coreDist(\mathbf{q}), dist(\mathbf{x}, \mathbf{q})) & \text{otherwise.} \end{cases}$$

For a core point $\mathbf{q}$, $reachDist(\mathbf{x}, \mathbf{q})$ is $dist(\mathbf{x}, \mathbf{q})$ if $\mathbf{x}$ is not belonging to $minPts-$NN of $\mathbf{q}$. Among several core points whose neighborhood contains $\mathbf{x}$, OPTICS seeks the *smallest* reachability distance $\mathbf{x}.reach$ for $\mathbf{x}$ from these core points. In other words, $\mathbf{x}.reach = \min_{\mathbf{q}_i} \{reachDist(\mathbf{x}, \mathbf{q}_i) \,|\, \mathbf{q}_i \text{ is core and } \mathbf{x} \in B_\varepsilon(\mathbf{q}_i)\}$.

OPTICS can be implemented as a nested loop as shown in Algorithm 6. In the outer loop (Line 4), a random $\mathbf{q} \in \mathbf{X}$ is selected and inserted into an empty cluster ordering $O$. If $\mathbf{q}$ is core, each point $\mathbf{x} \in B_\varepsilon(\mathbf{q})$ is inserted into a *priority queue* $Q$ with $reachDist(\mathbf{x}, \mathbf{q})$ as the key value. An inner loop (Line 12) that successively pops $\mathbf{q}'$ from $Q$ until $Q$ is empty. We can see that the priority of $\mathbf{q}' \in Q$ is determined by their smallest reachability distance w.r.t. current core points processed so far. The point with the smallest reachability distance in $Q$ is always popped first (due to the priority queue) and inserted into the ordering $O$.

We note that our presented OPTICS in Algorithm 6 is slightly different from the original version [17]. Since we do not know how to implement decrease-key operation efficiently on the priority queue [1], we propose a "lazy deletion" where we keep adding $\mathbf{x}$ into $Q$ with the new key. Though $\mathbf{x}$ might be duplicated on $Q$, by checking whether or not $\mathbf{x}$ is processed (Lines 16–18), we can process each point exactly once and output it into the cluster ordering $O$. When a point $\mathbf{x}$ is inserted into the ordering $O$, $\mathbf{x}.reach$ at the time it was popped out of $Q$ is recorded. Therefore, by setting $\varepsilon$ large enough, OPTICS outputs a cluster ordering that can be used as a visualization to extract clustering structure for smaller values of $\varepsilon$.

**Time and space complexity.** Similar to DBSCAN, the running time of OPTICS is $O(n^2)$ makes it impractical on large-scale data sets. Fast implementations of OPTICS with large values of $\varepsilon$ will require $O(n^2)$ memory to store the matrix distance between the core points and its neighborhood points. Such implementations are infeasible for large $n$.

## A.2 sOPTICS

To guide sDBSCAN's parameter setting, Algorithm 5 presents *sOPTICS*, an OPTICS variant with the identified core points and their neighborhood subsets provided by Algorithm 3. Given the identified core point $\mathbf{q}$ and $\widetilde{B}_\varepsilon(\mathbf{q})$, we store the set $\{dist(\mathbf{x}, \mathbf{q}) \text{ for each } \mathbf{x} \in \widetilde{B}_\varepsilon(\mathbf{q})\}$ (Line 16 of Alg. 3). Using this set, we can estimate $coreDist(\mathbf{q})$ as the $minPts$-NN distance between $\mathbf{q}$ and $\widetilde{B}_\varepsilon(\mathbf{q})$. Though such estimation is an *upper bound* of $coreDist(\mathbf{q})$, the tightness of the upper bound and the accurate neighborhood approximation make the reachability-plot provided by sOPTICS very similar to the

---

[1]C++ STL priority queue does not support decrease-key function.

---

**Algorithm 6** OPTICS

---

1: **Inputs**: $\mathbf{X}, \varepsilon, minPts$, the set $C = \{(\mathbf{q}, B_\varepsilon(\mathbf{q}), coreDist(\mathbf{q}),$
   $\{dist(\mathbf{x}, \mathbf{q})$ for each $\mathbf{x} \in B_\varepsilon(\mathbf{q})\}) \mid \mathbf{q}$ is core$\}$
2: Initialize an empty cluster ordering $O$
3: $\mathbf{q}.reach \leftarrow \infty$ for each $\mathbf{q} \in \mathbf{X}$
4: **for** each *unprocessed* $\mathbf{q} \in \mathbf{X}$ **do**
5:     Mark $\mathbf{q}$ as *processed*, and insert $\mathbf{q}$ into $O$
6:     **if** $\mathbf{q}$ is core **then**
7:         Seeds $\leftarrow$ empty priority queue $Q$
8:         **for** each $\mathbf{x} \in B_\varepsilon(\mathbf{q})$ **do**
9:             **if** $\mathbf{x}$ is *unprocessed* **then**
10:                 $\mathbf{x}.reach \leftarrow min(\mathbf{x}.reach, reachDist(\mathbf{x}, \mathbf{q}))$
11:                 Insert $(\mathbf{x}, \mathbf{x}.reach)$ into $Q$
12:         **while** $Q$ is not empty **do**
13:             $\mathbf{q}' \leftarrow Q.pop()$
14:             **if** $\mathbf{q}'$ is *processed* **then**
15:                 **continue**
16:             Mark $\mathbf{q}'$ as *processed*, and insert $\mathbf{q}'$ into $O$
17:             **if** $\mathbf{q}'$ is core **then**
18:                 **for** each *unprocessed* $\mathbf{x} \in B_\varepsilon(\mathbf{q}')$ **do**
19:                     $\mathbf{x}.reach \leftarrow min(\mathbf{x}.reach, reachDist(\mathbf{x}, \mathbf{q}'))$
20:                     Insert $(\mathbf{x}, \mathbf{x}.reach)$ into $Q$
21: **return** Cluster ordering $O$

---

original OPTICS with the same valley areas. Therefore, sOPTICS is a fast and reliable tool to guide DBSCAN and sDBSCAN parameter settings and to visualize the clustering structure.

Ensuring sOPTICS recovers OPTICS's result is difficult without any further assumptions as OPTICS's result is sensitive to the order of processed core points and their exact neighborhoods. Hence, we will discuss sOPTICS's quality in practical scenarios.

Like sDBSCAN, sOPTICS only considers top-$m$ closest/furthest points to any random vector where $m = O(minPts)$. Given a core point $\mathbf{q}$, the core distance of $\mathbf{q}$ derived from the set $\widetilde{B}_\varepsilon(\mathbf{q})$ is an *upper bound* of $coreDist(\mathbf{q})$. Since a core point $\mathbf{q}$ is often surrounded by many other core points in a dense cluster, $\widetilde{B}_\varepsilon(\mathbf{q})$ tends to contain core points. Hence, the upper bound of reachability distance provided by sOPTICS is tight. When the clustering structure is well separated, i.e., valleys in the reachability-plot are deep, sOPTICS with $m = O(minPts)$ is a reliable tool to guide the selection of $(\varepsilon, minPts)$ for DBSCAN. Importantly, the extra space of sOPTICS is linear, i.e. $O(nk \cdot minPts)$, as the approximate neighborhood size of each point is $O(k \cdot minPts)$.

**Time complexity.** Similar to sDBSCAN, sOPTICS runs in $O(dn \cdot minPts + nD \log(D))$ time for preprocessing and identifying core points and its approximate neighborhood. For each point $\mathbf{q}$, sOPTICS keeps $|\widetilde{B}_\varepsilon(\mathbf{q})| = O(k \cdot minPts)$ points and distance values. The size of the priority queue $Q$ of sOPTICS is $O(nk \cdot minPts)$. Therefore, for a constant $k$, sOPTICS runs in $O(dn \cdot minPts + nD \log(D) + n \cdot minPts \cdot \log(n \cdot minPts))$. When $D = o(n)$, sOPTICS run in *subquadratic* time.

**Space complexity.** sOPTICS needs $O(nk + D \cdot minPts)$ extra space to store $O(k)$ closest/furthest vectors for each point, and $O(minPts)$ points closest/furthest to each random vector. Due to the priority queue of size $O(nk \cdot minPts)$, sOPTICS uses $O(nk + D \cdot minPts + nk \cdot minPts)$ extra space, which is linear when $D = o(n)$, and *independent* on any value of $\varepsilon$. Note that scikit-learn uses $O(n^2)$ memory and hence could not run OPTICS on million-point data sets.

# B  Proof of Lemma 1, Lemma 3, and extension to other distance measures

## B.1  Proof of Lemma 1

We briefly describe the proof of Lemma 1 where points are on the unit sphere $\mathcal{S}^{d-1}$. We note that Lemma 1 holds on general Euclidean space as detailed in [22].

Given two vectors $\mathbf{q}, \mathbf{x} \in \mathcal{S}^{d-1}$ and any random Gaussian vector $\mathbf{r}_i \in \mathbb{R}^d$, we let $Q_i = \mathbf{q}^\top \mathbf{r}_i$ and $X_i = \mathbf{x}^\top \mathbf{r}_i$. It is well known that $Q_i \sim N(0,1), X_i \sim N(0,1)$, and $Q_i$ and $X_i$ are normal bivariates from $N(0,0,1,1,\rho)$ where $\rho = \mathbf{x}^\top \mathbf{q}$. Let $(Q_1, X_1), (Q_2, X_2), \ldots, (Q_D, X_D)$ be $D$ random samples from $N(0,0,1,1,\rho)$ generated by $D$ Gaussian vectors $\mathbf{r}_i, 1 \leq i \leq D$. We form the concomitants of normal order statistics by descendingly sorting these pairs based on $Q$-value.

Let $\mathbf{r}_* = \arg\max_{\mathbf{r}_i} \mathbf{q}^\top \mathbf{r}_i$. Let $X_{[1]}$ be the concomitant of the first (maximum) order statistic $Q_{(1)}$. Then we have $Q_{(1)} = \mathbf{q}^\top \mathbf{r}_*, X_{[1]} = \mathbf{x}^\top \mathbf{r}_*$. The theory of concomitants of extreme order statistics [18, 32] studies the asymptotic behavior of the concomitants $X_{[1]}$ given the asymptotic behavior of $Q_{(1)}$ when $D$ goes to infinity.

The seminal work of David and Galambos [18] establishes the following properties of concomitants of normal order statistics.

$$\mathbf{E}\left[Q_{(1)}\right] \xrightarrow{D} \sqrt{2\ln(D)}, \ \mathbf{Var}\left[Q_{(1)}\right] \xrightarrow{D} 0,$$

$$\mathbf{E}\left[X_{[1]}\right] = (\mathbf{x}^\top \mathbf{q})\mathbf{E}\left[Q_{(1)}\right] \xrightarrow{D} \mathbf{x}^\top \mathbf{q}\sqrt{2\ln(D)},$$

$$\mathbf{Var}\left[X_{[1]}\right] = 1 - (\mathbf{x}^\top \mathbf{q})^2 + (\mathbf{x}^\top \mathbf{q})^2 \mathbf{Var}\left[Q_{(1)}\right] \xrightarrow{D} 1 - (\mathbf{x}^\top \mathbf{q})^2,$$

$$X_{[1]} \xrightarrow{D} N\left(\mathbf{E}\left[X_{[1]}\right], \mathbf{Var}\left[X_{[1]}\right]\right).$$

Since Gaussian distribution is symmetric, we can use both $X_{[1]}$ and $-X_{[D]}$ corresponding to the maximum $Q_{(1)}$ associated to the closest random vector and minimum $Q_{(D)}$ associated to the furthest random vector to $\mathbf{q}$ for estimating $\mathbf{x}^\top \mathbf{q}$. This proves the claim of Lemma 1.

## B.2   Proof of Lemma 3

Given $D$ random vectors $\mathbf{r}_i \in \mathbb{R}^d$, $i \in [D]$, whose coordinates are randomly selected from the standard normal distribution $N(0,1)$, and the sign function $\mathrm{sgn}(\cdot)$, we randomly projects $\mathbf{x}$ and $\mathbf{q}$ onto these $D$ Gaussian random vectors. For significantly large $D$ random vectors $\mathbf{r}_i$, w.l.o.g. we assume that $\mathbf{r}_1 = \arg\max_{\mathbf{r}_i} |\mathbf{q}^\top \mathbf{r}_i|$, we have

$$\mathbf{x}^\top \mathbf{r}_1 \sim N\left(\mathrm{sgn}(\mathbf{q}^\top \mathbf{r}_1) \cdot \mathbf{x}^\top \mathbf{q}\sqrt{2\ln(D)}, 1 - (\mathbf{x}^\top \mathbf{q})^2\right).$$

For a constant $k > 0$, this property (i.e. Lemma 1) also holds for the top-$k$ closest/furthest vectors to $\mathbf{q}$ due to the asymptotic property of extreme normal order statistics [18, 22].

Given a core point $\mathbf{q}$, we denote by $\mathbf{x} \in B_\varepsilon(\mathbf{q})$ and $\mathbf{y} \in \mathbf{X} \setminus B_\varepsilon(\mathbf{q})$ any close and far away points to $\mathbf{q}$ regarding the distance threshold $\varepsilon$. W.l.o.g., we assume that $\mathbf{r}_1 = \arg\max_{\mathbf{r}_i} |\mathbf{q}^\top \mathbf{r}_i| = \arg\max_{\mathbf{r}_i} \mathbf{q}^\top \mathbf{r}_i$. Let $X = \mathbf{x}^\top \mathbf{r}_1, Y = \mathbf{y}^\top \mathbf{r}_1$ be random variable corresponding to the projection values of $\mathbf{x}, \mathbf{y}$ on $\mathbf{r}_1$, respectively. Then we have

$$X \sim N\left(\mathbf{x}^\top \mathbf{q}\sqrt{2\ln(D)}, 1 - (\mathbf{x}^\top \mathbf{q})^2\right), \ Y \sim N\left(\mathbf{y}^\top \mathbf{q}\sqrt{2\ln(D)}, 1 - (\mathbf{y}^\top \mathbf{q})^2\right).$$

Let $\alpha_{\mathbf{xy}} = \left(\mathbf{x}^\top \mathbf{q} - \mathbf{y}^\top \mathbf{q}\right) / \left(\sqrt{1 - (\mathbf{x}^\top \mathbf{q})^2} + \sqrt{1 - (\mathbf{y}^\top \mathbf{q})^2}\right)$, applying Chernoff bound [33] on the Gaussian variable $Y - X$ gives

$$\mathbf{Pr}\left[Y \geq X\right] \leq D^{-\left(\mathbf{x}^\top \mathbf{q} - \mathbf{y}^\top \mathbf{q}\right)^2 / \left(\sqrt{1-(\mathbf{x}^\top \mathbf{q})^2} + \sqrt{1-(\mathbf{y}^\top \mathbf{q})^2}\right)^2} = D^{-\alpha_{\mathbf{xy}}^2}.$$

Let $\alpha_* = \arg\min_{\mathbf{x} \in B_\varepsilon(\mathbf{q}), \mathbf{y} \in \mathbf{X} \setminus B_\varepsilon(\mathbf{q})} \alpha_{\mathbf{xy}}$ and $D = n^{2/\alpha_*^2}$, applying the union bound, we have

$$\mathbf{Pr}\left[\mathbf{x} \text{ is ranked higher than all } \mathbf{y} \in \mathbf{X} \setminus B_\varepsilon(\mathbf{q}) \text{ on } \mathbf{r}_1\right] \geq 1 - 1/n.$$

Since Lemma 1 holds for $k$ closest/furthest random vectors $\mathbf{r}_i$ to $\mathbf{q}$ due to their asymptotic independence, let $\mathbf{R}_\mathbf{q}$ be the set of these vectors associated to $\mathbf{q}$. By setting $D = n^{1/k\alpha_*^2}$ we have

$$\mathbf{Pr}\left[\mathbf{x} \text{ is ranked higher than all } \mathbf{y} \in \mathbf{X} \setminus B_\varepsilon(\mathbf{q}) \text{ on one vector in } \mathbf{R}_\mathbf{q}\right] \geq 1 - 1/n.$$

Assume that the event any point $\mathbf{x} \in B_\varepsilon(\mathbf{q})$ is ranked higher than all $\mathbf{y} \in \mathbf{X} \setminus B_\varepsilon(\mathbf{q})$ is independent, we have

$\mathbf{Pr}\left[\text{At least } minPts \text{ points in } B_\varepsilon(\mathbf{q}) \text{ appear on top-}minPts \text{ points associated to } \mathbf{R}_\mathbf{q}\right] \geq (1-1/n)^{minPts}.$

This proves the claim of Lemma 3.

### B.3 Extension to other similarity measures with random kernel features

We show how to extend sDBSCAN and sOPTICS to other popular distance measures, including L2, L1, $\chi^2$, and Jensen-Shannon (JS). We will utilize the random features [19, 20] to embed these distances into cosine distance. In particular, we study fast randomized feature mapping $f : \mathbb{R}^d \mapsto \mathbb{R}^{d'}$ such that $\mathbf{E}\left[f(\mathbf{x})^\top f(\mathbf{q})\right] = K(\mathbf{x}, \mathbf{q})$ where $K$ is the kernel function. We study Gaussian, Laplacian, $\chi^2$, and JS kernels as their randomized mappings are well-studied and efficiently computed. Also, the embeddings' extra costs are negligible compared to those of sDBSCAN and sOPTICS.

Given $\mathbf{x} = \{x_1, \ldots, x_d\}, \mathbf{y} = \{y_1, \ldots, y_d\}$ and $\sigma > 0$, the following are the definitions of the investigated kernels.

$$K_L(\mathbf{x}, \mathbf{y}) = e^{-\|\mathbf{x}-\mathbf{y}\|_1/\sigma} \; ; \; K_G(\mathbf{x}, \mathbf{y}) = e^{-\|\mathbf{x}-\mathbf{y}\|_2^2/2\sigma^2} \; ;$$

$$K_{\chi^2}(\mathbf{x}, \mathbf{y}) = \sum_i \frac{2x_i y_i}{x_i + y_i} \; ;$$

$$K_{JS}(\mathbf{x}, \mathbf{y}) = \sum_i \frac{x_i}{2} \log\left(\frac{x_i + y_i}{x_i}\right) + \frac{y_i}{2} \log\left(\frac{x_i + y_i}{y_i}\right).$$

We present here the random Fourier embeddings [19] for $K_L$ and $K_G$ with $\sigma = 1$. We first generate $d'$ random vectors $\mathbf{w}_i, i \in [d']$ whose coordinates are from $N(0, 1)$ for $K_G$ and $Lap(0, 1)$ for $K_L$. Our randomized mappings are:

$$f(\mathbf{x}) = \frac{1}{\sqrt{d'}}\{\sin(\mathbf{w}_i^\top \mathbf{x}), \cos(\mathbf{w}_i^\top \mathbf{x}) \,|\, i \in [d']\} \in \mathcal{S}^{2d'-1} \; .$$

Since $|f(\mathbf{x})^\top f(\mathbf{y})| \leq \|f(\mathbf{x})\|_2 \|f(\mathbf{y})\|_2 = 1$, the Hoeffding's inequality guarantees: $\mathbf{Pr}\left[|f(\mathbf{x})^\top f(\mathbf{y}) - K(\mathbf{x}, \mathbf{y})| \geq \delta\right] \leq e^{-d' \cdot \delta^2/2}$. By selecting $d' = O(\log(n))$, the randomized mapping $f$ preserves well the kernel function of every pair of points. Hence, sDBSCAN and sOPTICS on these randomized embeddings output similar results to DBSCAN and OPTICS on the corresponding distance measures.

**Complexity.** For $K_G, K_L$, the embedding time is $O(dd')$. For $K_{\chi^2}, K_{JS}$, the embedding time is $O(d')$ for $d'$ random features by applying the sampling and scaling approaches [20]. Empirically, the random feature construction time is negligible compared to the sDBSCAN and sOPTICS time. As we execute random projections on the constructed random features for each point and compute $dist(\mathbf{x}, \mathbf{y})$ using the original data, we only need a small extra space to store random vectors $\mathbf{w}_i$.

## C   Additional experiments

We use the Eigen library [2] for SIMD vectorization on computing the distances. It will only give advantages on dense data sets, e.g., Pamap2. Our sDBSCAN and sOPTICS are multi-threading friendly. We only add `#pragma omp parallel` directive on the `for` loop when preprocessing and finding the neighborhood for each point.

**Clustering competitors.** We tried several clustering algorithms on scikit-learn, including DBSCAN, spectral clustering, and kernel $k$-means. They could not work on million-point data sets given our DRAM of 128 GB. The released JAVA implementation of random projection-based DBSCAN [13] cannot run even on the small Mnist data set.

Our competitors include pDBSCAN [6] [3] as a representative grid-based approach, DBSCAN++ [14] [4] and sngDBSCAN [15] [5] as representatives for sampling-based approaches. DBSCAN++ has two variants, including DBSCAN++ with uniform initialization (called uDBSCAN++) that uses KD-Trees to speed up the search of core points and $k$-center initialization (called kDBSCAN++). We also compare with multi-threading scikit-learn implementations of DBSCAN and OPTICS [6].

---

[2] https://eigen.tuxfamily.org

[3] https://sites.google.com/view/approxdbscan

[4] https://github.com/jenniferjang/dbscanpp

[5] https://github.com/jenniferjang/subsampled_neighborhood_graph_dbscan

[6] https://scikit-learn.org/stable/modules/clustering.html

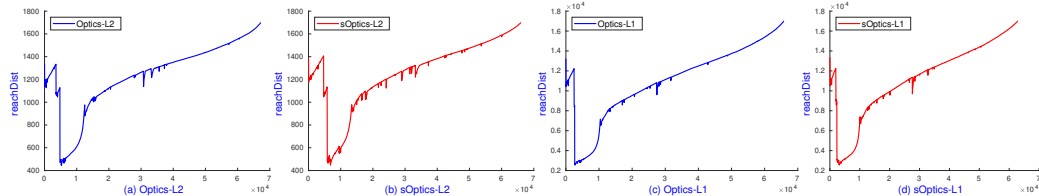

Figure 4: Reachability-plot dendrograms of OPTICS and sOPTICS over L2 and L1 on Mnist. While sOPTICS needs less than **30 seconds**, scikit-learn OPTICS requires **1.5 hours** on L2 and **0.5 hours** on L1.

Since the current sngDBSCAN only supports L2 and single thread, we re-implement sngDBSCAN to support L1, $\chi^2$, and JS distance with multi-threading.

**Datasets.** We conduct experiments on three popular data sets: Mnist ($n = 70,000, d = 784$, # clusters = 10), Pamap2 ($n = 1,770,131, d = 51$, # clusters = 18), and Mnist8m ($n = 8,100,000, d = 784$, # clusters = 10), as shown in Table 2. For Pamaps2, we discarded instances that contain NaN values and removed the dominated class 0 corresponding to the transient activities. We note that Mnist and Mnist8m are sparse data sets with at least 75% sparsity, while Pamap2 is dense.

### C.1 sOPTICS vs. scikit-learn OPTICS on Mnist over L1 and L2

This subsection measures the performance of sOPTICS and scikit-learn OPTICS with multi-threads. We show that sOPTICS can output the same OPTICS's results in less than *30 seconds*, which is up to $180\times$ speedups over scikit-learn OPTICS.

We use scikit-learn OPTICS to generate the dendrogram given $minPts = 50$. To select a reasonable value of $\varepsilon$ for OPTICS to reduce the running time, we randomly sample 100 points and use the average top-100 nearest neighbor distances of these sampled points as $\varepsilon$. Accordingly, we use $\varepsilon = 1,800$ for L2 and $\varepsilon = 18,000$ for L1.

To recover OPTICS's results, sOPTICS uses $D = 2,048, k = 2$ and $m = 1,000$. For L2 and L1 distances, sOPTICS requires two additional parameters, including the scale $\sigma$ and the number of embeddings $d'$, for random Fourier embeddings. We observe that the performance of sOPTICS is insensitive to $\sigma$ and $d'$, hence we simply set $\sigma = 2 * \varepsilon$ and $d' = 1,024$ for our experiments.

We run both scikit-learn OPTICS and sOPTICS with 64 threads. Regarding speed, sOPTICS needs less than 30 seconds, while OPTICS takes hours to finish. This shows the advantages of sOPTICS in accurately finding candidates to construct neighborhoods and efficiently utilizing multi-threading. Regarding accuracy, sOPTICS outputs nearly the same graphs as OPTICS, as shown in Figure 4. Since there are more valleys on L2 than L1, the clustering accuracy regarding the ground truth using L2 will be higher than L1.

### C.2 sDBSCAN vs. scikit-learn DBSCAN on Mnist over cosine

We first normalize all data points. Since cosine distance is identical to L2 on a unit sphere, we use scikit-learn DBSCAN on L2 to compare with sDBSCAN on cosine distance. Given $minPts = 50$, we set $\varepsilon = 0.11$ as DBSCAN returns the highest NMI of 43% with the ground truth. sDBSCAN has two main parameters on cosine distance: top-$m$ points closest/furthest to a random vector and top-$k$ closest/furthest random vectors to a point. These parameters govern the accuracy and efficiency of sDBSCAN. We set $D = 1,024$ and vary $m, k$ in the next experiments.

Table 2: The data sets

| Datasets | $n$ | $d$ | # clusters |
|---|---|---|---|
| Mnist | 70,000 | 784 | 10 |
| Pamap2 | 1,770,131 | 51 | 18 |
| Mnist8m | 8,100,000 | 784 | 10 |

**Accuracy and efficiency of sDBSCAN.** For sDBSCAN, we vary $k = \{40, 20, \ldots, 1\}$, $m = \{50, 100, \ldots, 2,000\}$ such that it computes nearly the same $2km$ distances for each point. Table 3 shows the performance of multi-threading sDBSCAN on DBSCAN's outputs with $\varepsilon = 0.11, minPts = 50$. We can see that sDBSCAN can recover DBSCAN's result with larger $m$. These findings justify our theoretical result as larger $m$ increases the chance of examining all points in the set $S_i$ and $R_i$ corresponding to the random vector $\mathbf{r}_i$, increasing the chance to recover the DBSCAN's output.

In the last column with $m = 2,000$, sDBSCAN with $k = 2$ achieves NMI of 95% but runs 2 times slower than $k = 1$ since it nearly doubles the candidate size. sDBSCAN with all configurations runs at least $3\times$ faster than scikit-learn DBSCAN. This presents the advantages of sDBSCAN in accurately finding candidates to find core points and efficiently utilizing the multi-threading architecture.

**sDBSCAN's parameter setting.** Given $m \geq minPts$ and $k$, each point needs at most $2km$ distance computations due to the duplicates on top-$m$ candidates corresponding to $2k$ investigated random vectors. As a larger $k$ leads to more duplicates among $2km$ candidates, given a fixed budget $B = 2km$, Table 3 shows that $k = 1, m = B/2$ has higher accuracy but significantly higher running time than $k = k_0 > 1, m = B/2k_0$.

While larger $m$ and $D$ tend to increase the accuracy of sDBSCAN, they affect space and time complexity. Since we often set $D = 1,024$ to ensure Lemma 3 holds, and the memory resource is limited due to large data sets, we set $m = minPts = 50$ and $k = \{5, 10\}$ for most experiments. Compared to the ground truth, this setting does not affect the accuracy of sDBSCAN but uses significantly less computational resources than other configurations.

**Running time of sDBSCAN's components with multi-threading.** Given $\varepsilon = 0.11$, $m = minPts = 50$, $k = 5$, Table 4 shows the running time of sDBSCAN components on 1 and 64 threads. On 1 thread, we can see that finding neighborhoods is the primary computational bottleneck while forming clusters is negligible. This is due to the expensive random access operations to compute $2km$ distances for each point. As the main components of sDBSCAN, including preprocessing (Alg. 2) and finding neighborhoods (Alg. 3) can be sped up with multi-threading, we achieve nearly $10\times$ speedup with 64 threads. We also observe that the construction time of the randomized embeddings to support L2, L1, $\chi^2$, and JS is similar to the preprocessing time and is negligible to that of finding neighborhoods.

### C.3   sOPTICS graphs on Pamap2 and Mnist8m

We use the average top-100 nearest neighbor distances of 100 sampled points to select a suitable $\varepsilon$ for sOPTICS. Figure 5 and 6 show sOPTICS's graphs of Pamap2 and Mnist8m with $minPts = 50$. As Pamap2 contains negative features, we only run on cosine, L2, and L1 distances. sOPTICS uses $D = 1,024$, $\sigma = 2 * \varepsilon$, $d' = 1,024$ runs in less than *2 minutes* and *11 minutes* on Pamap2 and Mnist8m, respectively. Both figures show that L2 is less relevant than the other distances as it does not show clear valley areas. This reflects the need to use other distance measures rather than L2 for density-based clustering to achieve reasonable performance.

Table 3: Comparison of sDBSCAN with the DBSCAN's output on cosine distance with $\varepsilon = 0.11, minPts = 50$ over different $k$ and $m$ on Mnist. The scikit-learn DBSCAN runs in **71 seconds**.

| $m$ | 50 | 100 | 200 | 400 | 1,000 | 2,000 | **2,000** |
|---|---|---|---|---|---|---|---|
| $k$ | 40 | 20 | 10 | 5 | 2 | 1 | **2** |
| NMI | 69% | 76% | 81% | 86% | 88% | 91% | **95%** |
| Time (s) | 8.8 | 9.9 | 10.5 | 11.1 | 12.1 | 12.6 | 23 |

Table 4: Running time of sDBSCAN components in seconds with $D = 1,024, k = 5, m = minPts = 50, \varepsilon = 0.11$ on Mnist.

| # Threads | Preprocess | Find core points | Cluster | Total |
|---|---|---|---|---|
| 1 thread | 1.424 | 7.745 | 0.001 | 9.198 |
| 64 threads | 0.160 | 0.700 | 0.001 | 0.862 |

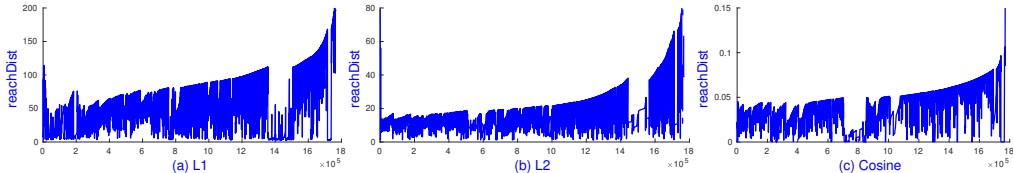

Figure 5: sOPTICS's graphs on L1, L2 and cosine distances on Pamap2. Each runs within **2 minutes**. scikit-learn OPTICS could not finish in **4 hours**.

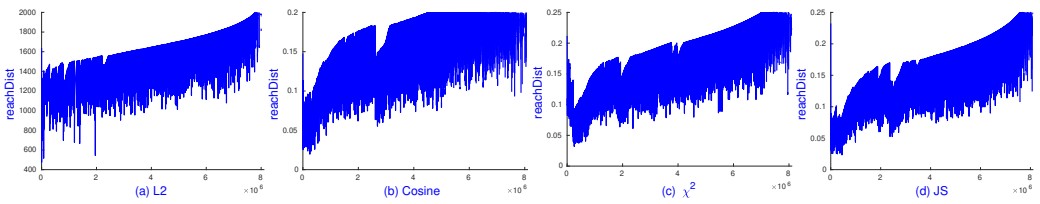

Figure 6: sOPTICS's graphs on L2, cosine, $\chi^2$, and JS distances on Mnist8m. Each runs within **11 minutes**. scikit-learn OPTICS could not finish in **4 hours**.

## C.4 Detailed comparison on Pamap2

As L2 is inferior to L1 and cosine on sampling-based DBSCAN, Table 5 reports the NMI scores and running time of studied algorithms on L1 and cosine distances over a larger range of $\varepsilon$. While sDBSCAN shows a low NMI on cosine, its accuracy is 45% on L1, which is higher than sngDBSCAN on both cosine and L1. As sngDBSCAN uses $p = 0.01$, each point will compute $np \sim 17,700$ distances compared to $2km = 1000$ of sDBSCAN. Hence, sDBSCAN runs up to $14\times$ faster than sngDBSCAN on L2. On L1, sDBSCAN is nearly $100\times$ faster than DBSCAN. By changing $k = 5, m = 200$, sDBSCAN reaches 48% NMI on L1 and still runs in less than *1 minute*. Compared

Table 5: The NMI on the best $\varepsilon$ and running time comparison on cosine and L1 distances on Pamap2. The upper 3 algorithms run in multi-threading with $10\times$ speedup compared to 1 thread while the lower ones use 1 thread.

| Algorithms | Cosine ($\varepsilon \in [0.005, 0.05]$) | | | L1 ($\varepsilon \in [30, 80]$) | | |
|---|---|---|---|---|---|---|
| | NMI | Time | $\varepsilon$ | NMI | Time | $\varepsilon$ |
| DBSCAN | **47%** | 28.4 min | 0.005 | **50%** | 29.3 min | 40 |
| sDBSCAN | 34% | **0.2 min** | 0.04 | **46%** | **0.3 min** | 60 |
| sngDBSCAN | 42% | 2.8 min | 0.015 | 40% | 2.7 min | 50 |
| uDBSCAN++ | **46%** | 3 min | 0.015 | – | – | – |
| kDBSCAN++ | 39% | 13.4 min | 0.05 | – | – | – |
| $k$-mean++ ($k = 18$) | 36% | 0.4 min | – | – | – | – |

Table 6: The NMI on the best $\varepsilon$ and running time comparison of multi-threading DBSCAN variants on L2 and cosine on Mnist8m. Kernel $k$-means ($k = 10$) [21] runs in **15 minutes** on a supercomputer of 32 nodes and achieves NMI 41%.

| Algorithms | L2 ($\varepsilon \in [1100, 1500]$) | | | Cosine ($\varepsilon \in [0.1, 0.2]$) | | |
|---|---|---|---|---|---|---|
| | NMI | Time | $\varepsilon$ | NMI | Time | $\varepsilon$ |
| sDBSCAN-1NN | **36%** | 8 min | 1400 | **37%** | 14 min | 0.16 |
| sDBSCAN | 25% | **7 min** | 1350 | 32% | **8 min** | 0.16 |
| sngDBSCAN ($p = 0.001$) | 22% | 43 min | 1150 | 26% | 42 min | 0.16 |

to $k$-mean++, multi-threading sDBSCAN runs faster and offers significantly higher NMI with L1. We emphasize that the advantage of multi-threading sDBSCAN comes from its simplicity, as it is effortless engineering to run sDBSCAN on multi-threads.

Among sampling-based approaches, uDBSCAN++ gives the highest NMI score of 46%, while kDBSCAN++ is very slow. kDBSCAN++ runs approximately $5\times$ slower than uDBSCAN due to the overhead of $k$-center initialization and the efficiency of KD-Trees on Pamap2 with $d = 51$. In general, sDBSCAN provides competitive clustering accuracy and runs significantly faster than other DBSCAN variants. Like Mnist, our sOPTICS on Pamap2 runs in less than *2 minutes*, even faster than any sampling-based implementations.

## C.5 Detailed comparison on Mnist8m

As Mnist8m is non-negative, we conduct experiments on cosine, L2, and L1, $\chi^2$, and JS distances. Similar to Mnist, sOPTICS graphs show L1 is inferior, so we do not report L1 here.

uDBSCAN++ and kDBSCAN++ with $p = 0.001$ could not finish after 3 hours. It is not surprising as our multi-threading sngDBSCAN runs in nearly 1 hour, and single thread sngDBSCAN is at least $4\times$ faster than DBSCAN++ variants on Mnist. As we cannot run scikit-learn $k$-means++, we use the result of 41% NMI given by a fast kernel $k$-means [21] running on a supercomputer with 32 nodes, each of which has two 2.3GHz 16-core (32 threads) Haskell processors and 128GB of DRAM. This configuration of a *single* node is similar to our conducted machine.

Table 6 and 7 summarize the performance of multi-threading sDBSCAN-1NN, sDBSCAN, and sngDBSCAN, including the NMI score on the best $\varepsilon$ and the running time, on L2, cosine, $\chi^2$, and JS. sDBSCAN-1NN runs in at most 15 minutes and returns the highest NMI among DBSCAN variants with a peak of 38% NMI on $\chi^2$ and JS. We emphasize that kernel $k$-means with Nyström approximation [21, Table 4] also needs 15 minutes on a supercomputer and gives 41% NMI.

sDBSCAN runs faster than sDBSCAN-1NN as it does not assign labels to non-core points, and still achieves significantly higher NMI scores than sngDBSCAN on all 4 studied distances. Among 4 distances, L2 shows lower accuracy but runs faster than $\chi^2$ and JS due to the faster distance computation. sDBSCAN runs $6.4\times$ faster than sngDBSCAN, which is justified by the number of distance computations $np = 8,100$ of sngDBSCAN compared to $2km = 1000$ of sDBSCAN. As sDBSCAN-1NN samples $0.01n$ core points to build the approximate 1NN classifier, the running time overhead of this extra step is smaller than sDBSCAN's time.

## C.6 Sensitivity of parameters used in random kernel features

We first show empirical results on random kernel mappings that facilitate sDBSCAN and sOPTICS on L1, L2, $\chi^2$, and JS distance. We use Pamap2 for L1 and L2 as it contains negative values and Mnist for $\chi^2$ and JS as it does not contain negative values.

**L1 and L2 on Pamap2.** We carry out experiments to evaluate the sensitivity of the parameter $\sigma$ used on random kernel mappings on L1 and L2. We fix $k = 10, m = minPts = 50, D = 1024, d' = 1024$ and vary $\sigma$ for L1 and L2. Figure 7 shows the accuracy of sDBSCAN on the recommended range of $\varepsilon$ by sOPTICS graphs with $\sigma = \{50, 100, 200, 400\}$ for L1 on Figure 8 and $\sigma = \{20, 40, 80, 160\}$ for L2 on Figure 9.

Table 7: The NMI on the best $\varepsilon$ and running time comparison of multi-threading DBSCAN variants on $\chi^2$ and JS distances on Mnist8m. Kernel $k$-means ($k = 10$) [21] runs in **15 minutes** on a supercomputer of 32 nodes and achieves NMI 41%.

| Algorithms | $\chi^2$ ($\varepsilon \in [0.1, 0.2]$) | | | JS ($\varepsilon \in [0.1, 0.2]$) | | |
|---|---|---|---|---|---|---|
| | NMI | Time | $\varepsilon$ | NMI | Time | $\varepsilon$ |
| sDBSCAN-1NN | **38%** | 15 min | 0.17 | **38%** | 15 min | 0.15 |
| sDBSCAN | 32% | **10 min** | 0.17 | 31% | **10 min** | 0.15 |
| sngDBSCAN ($p = 0.001$) | 28% | 64 min | 0.15 | 27% | 64 min | 0.17 |

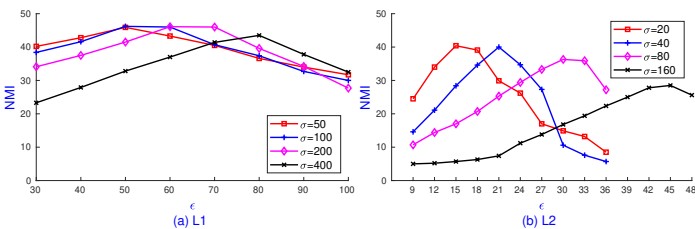

Figure 7: sDBSCAN's NMI on L1 and L2 with various $\sigma$ on Pamaps with $k = 10, m = minPts = 50$. Each runs in less than **20 seconds**.

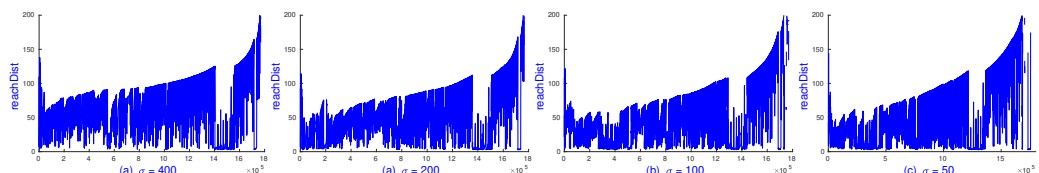

Figure 8: sOPTICS's graphs on L1 with various $\sigma$ on Pamaps with $k = 10, m = minPts = 50$. Each runs in less than **2 minutes**.

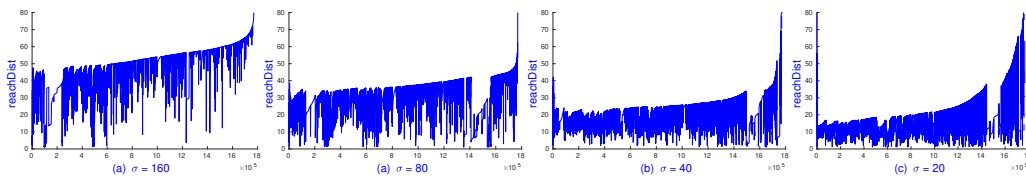

Figure 9: sOPTICS's graphs on L2 with various $\sigma$ on Pamaps with $k = 10, m = minPts = 50$. Each runs in less than **2 minutes**.

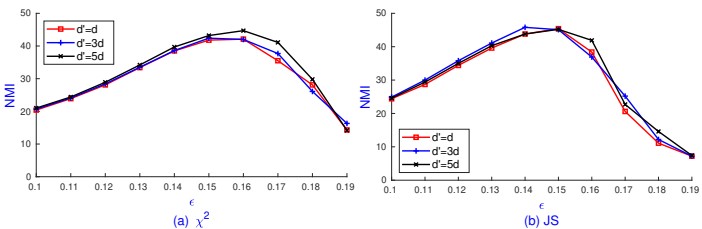

Figure 10: sDBSCAN's NMI on $\chi^2$ and JS with various $\sigma$ on Mnist with $k = 10, m = minPts = 50$. Each runs in less than **20 seconds**.

While L1 provides 46% NMI, higher than just 40% NMI by L2, the setting $\sigma = 2\varepsilon$ reaches the peak of NMI for both L1 and L2. For example, sDBSCAN at $\varepsilon = 50, \sigma = 100$ reaches NMI 46% for L1. sDBSCAN at $\varepsilon = 15, \sigma = 20$ and $\varepsilon = 21, \sigma = 40$ reach NMI of 40% for L2. Also, the performance of sDBSCAN is not sensitive to the value $\sigma$, especially with the guide of sOPTICS graphs. For example on L2 as shown in Figure 9, $\sigma = 160$ suggests the suitable range $[40, 50]$ while $\sigma = 80$ shows $[30, 40]$. The best values of $\varepsilon$ of these two $\sigma$ values are clearly on these ranges, as shown in Figure 7(b). This observations appear again on $\sigma = \{20, 40\}$ for the range of $[10, 20]$ and $[20, 30]$, respectively.

$\chi^2$ **and JS on Mnist.** We carry out experiments to evaluate the sensitivity of the parameter $d'$ used on random kernel mappings on $\chi^2$ and JS. We fix $k = 5, m = minPts = 50, D = 1024$ and vary $d'$. We note that [20] approximates $\chi^2$ and JS distance using the sampling approach, hence we set

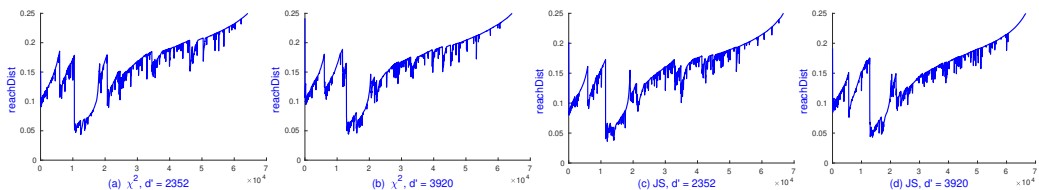

Figure 11: sOPTICS's graphs on $\chi^2$ and JS with $d' = \{3d, 5d\}$ on Mnist with $k = 5, m = minPts = 50$. Each runs in less than **3 seconds**.

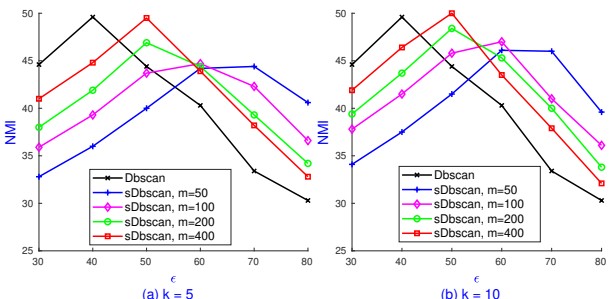

Figure 12: sDBSCAN's NMI on L1 with various $k, m$ on Pamaps with $minPts = 50$.

the sampling interval as 0.4, as suggested on scikit-learn [7]. We also note that $d'$ should be set as $(2l + 1)d$ for $l \in \mathbb{N}$. Hence, we set $d' = \{3d, 5d\}$ in our experiment.

Figure 11 shows sOPTICS graphs of $\chi^2$ and JS on $d' = \{3d, 5d\}$. They are very similar to the sOPTICS graphs on $d' = d$ where the range of $\varepsilon$ should be in $[0.15, 0.2]$ for $\chi^2$ and $[0.13, 0.18]$ for JS. Therefore, we can see that sDBSCAN with such a recommended range of $\varepsilon$ will reach the peak of accuracy.

Figure 10 shows that sDBSCAN reaches the peak at $\varepsilon = 0.16$ with $\chi^2$ and $\varepsilon = 0.14$ with JS. Using $d' = 5d$ slightly increases the accuracy compared to $d' = d$ though it is not significant on JS. Both measures offer the highest accuracy with 45% NMI.

### C.7 Sensitivities of *k, m* of sDBSCAN

We carry out experiments on Pamap2 using L1 to study the performance of sDBSCAN on various values of $k, m$ since L1 shows superiority compared to cosine and L2. We fix $\sigma = 200, D = 1024$ and consider the range of $\varepsilon \in [30, 80]$. As increasing $k, m$ will increase the memory and running time of sDBSCAN, we first fix $k = \{5, 10\}$ and then increase $m = \{50, 100, 200, 400\}$.

Figure 12 shows that for a fix $k$, increasing $m$ slightly increases the accuracy of sDBSCAN. In particular, sDBSCAN with $m = 400$ at $\varepsilon = 50$ reaches the accuracy of the exact DBSCAN of 50% NMI. Regarding the time, sDBSCAN with $m = 400$ needs *1.5 minutes* and *xxx mins* for $k = 5$ and $k = 10$, respectively, which is significantly faster than *29.3 minutes* required by the exact DBSCAN.

### C.8 Neighborhood size *minPts = 100*

We present experiments on the setting of $minPts = 100$. We follow the same procedure that plots sOPTICS graphs first and use them to select the relevant range values of $\varepsilon$. We all use $k = 10, m = 100, D = 1024$ for Pamap2 and Mnist8m.

**Pamap2.** Figure 13 shows sOPTICS graphs on Pamap2 with L1, L2, and cosine distances where we use $\sigma = 200$ for L1, and $\sigma = 20$ for L2. It shows again that L2 is inferior than L1 and cosine

---

[7]https://github.com/scikit-learn/scikit-learn/blob/5c4aa5d0d/sklearn/kernel_approximation.py

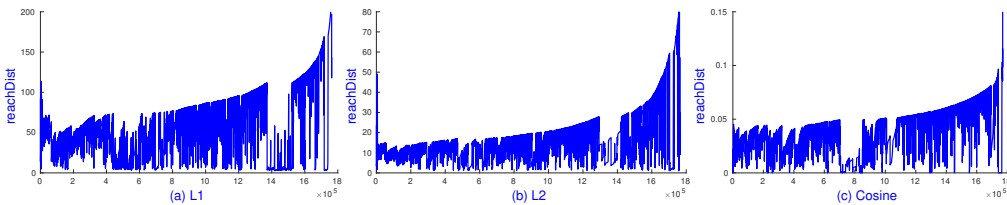

Figure 13: sOPTICS's graphs on L1, L2, cosine on Pamap2 with $k = 10, m = minPts = 100$. Each runs in less than **5 minutes**.

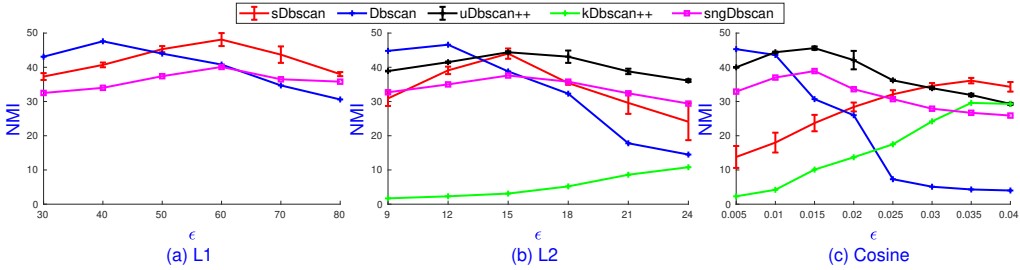

Figure 14: sDBSCAN's NMI on L1, L2, and cosine on Pamap2 with $k = 10, m = minPts = 100$. Each runs within **1 minute**.

distances. Figure 14 shows the accuracy of sDBSCAN compared to other competitors on Pamap2 with studied distances over the range of $\varepsilon$ suggested by their corresponding sOPTICS graphs.

SDBSCAN reaches the highest NMI at the suggested range of $\varepsilon$ by their sOPTICS graphs. In contrast, sampling-based approaches have to investigate a much wider range of $\varepsilon$ to achieve good performance. We observe that the performance of sDBSCAN is stable regarding $minPts$ though a larger $minPts$ requires larger $m$ which increases the running time. Each instance of sDBSCAN runs in 0.5 minutes, which is significantly faster than 3 minutes by uDBSCAN++ and 13.4 minutes by kDBSCAN++. sDBSCAN shows the advantage of running on many distance measures, which leads to the highest of 48% of NMI of L1. In contrast, uDBSCAN++ achieves the highest of 46% of NMI among cosine and L2 distances.

**Mnist8m.** Figure 15 shows sOPTICS graphs on Mnist8m with L2, cosine, $\chi^2$ and JS distances where we use $\sigma = 4000$ for L2, and $d' = d$ on $\chi^2$ and JS. It shows the advantages of sDBSCAN on supporting many distances and predicts that cosine, $\chi^2$ and JS provide higher accuracy than L2.

Figure 16 shows the accuracy of sDBSCAN variants compared to other competitors on Mnist8m with studied distances over the range of $\varepsilon$ suggested by their corresponding sOPTICS graphs. We consider *sDBSCAN-1NN* with the approximation 1NN heuristic to cluster border and noisy points detected by sDBSCAN. It shows again that sDBSCAN variants outperform sngDBSCAN on all studied distances. sDBSCAN-1NN on a single computer nearly reaches the accuracy of kernel $k$-means [21]. Regarding the speed, each instance of sDNSCAN-1NN runs in less than 15 minutes, which is the time requirement for kernel $k$-means on a supercomputer.

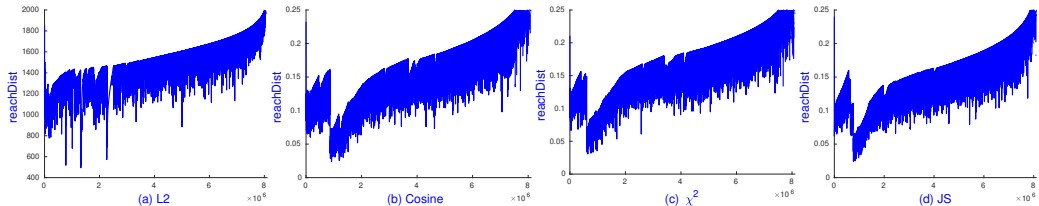

Figure 15: sOPTICS's graphs on L2, cosine, $\chi^2$, and JS distances on Mnist8m with $k = 10, m = minPts = 100$. Each runs in less than **20 minutes**.

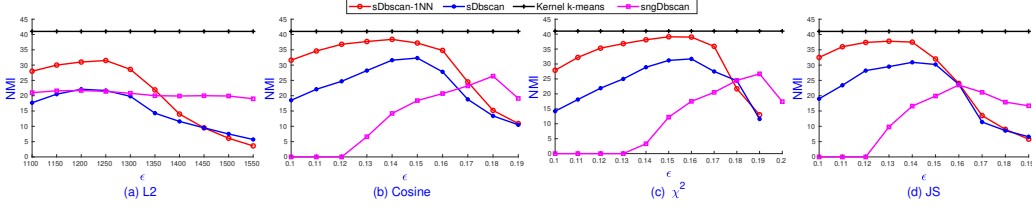

Figure 16: sDBSCAN's NMI on L2, cosine, $\chi^2$, and JS on Mnist8m with $k = 10, m = minPts = 100$. Each runs within **20 minutes**.

## C.9 Clustering accuracy with NMI and CC on Mnist and Pamap2

Figures 17 and 18 show the clustering accuracy with NMI and CC on Mnist and Pamap2 of sDbscan, DBSCAN, uDBSCAN++, kDBSCAN++, sngDBSCAN on $minPts = 50$. These figures show that sDBSCAN is competitive to other DBSCAN variants on different clustering accuracy measures.

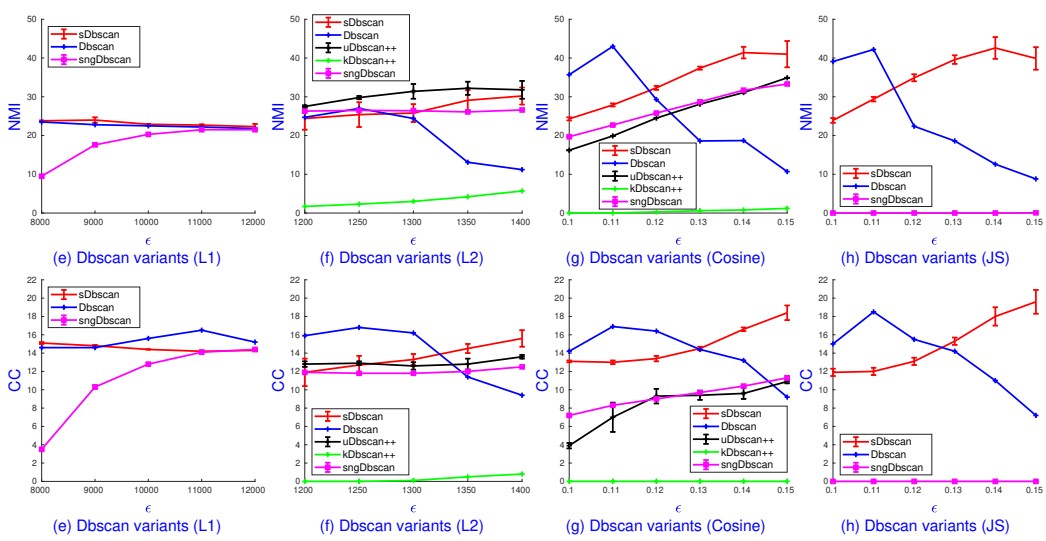

Figure 17: NMI and CC of DBSCAN variants on Mnist with L1, L2, cosine, JS over the range of $\varepsilon$ suggested by sOPTICS. Cosine and JS give the highest clustering accuracy.

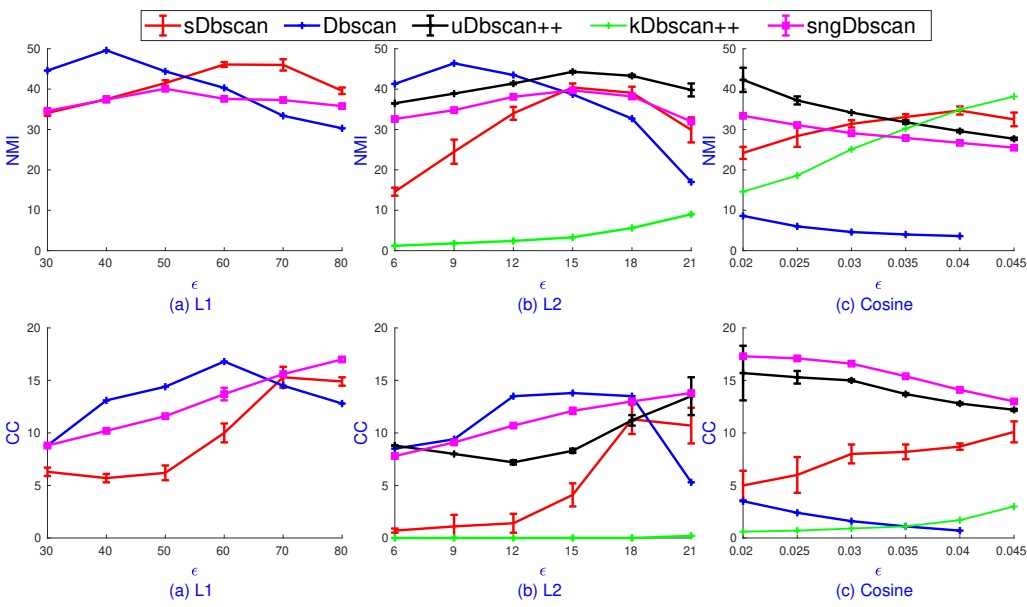

Figure 18: NMI and CC of DBSCAN variants on Pamap2 with L1, L2, cosine over the range of $\varepsilon$ suggested by sOPTICS. L1 gives the highest clustering accuracy.

