# OpenReview forum: "Scalable DBSCAN with Random Projections"
_NeurIPS.cc/2024/Conference — NeurIPS 2024 poster_

### Official Review · Reviewer_KGU7 · 2024-07-04

**Soundness:** 2
**Presentation:** 2
**Contribution:** 1
**Rating:** 4
**Confidence:** 5

**Summary:**

The authors presented a significant advancement in density-based clustering algorithms with the introduction of sDBSCAN. The scalability and speed improvements are particularly noteworthy, making it suitable for large datasets where traditional DBSCAN variants struggle. The theoretical underpinning provided confidence in the algorithm’s ability to maintain the clustering structure, while the empirical results demonstrate practical benefits. However, the paper would benefit from a deeper exploration of the algorithm's parameter sensitivity and a more detailed analysis of memory usage. The implementation complexity due to random projections and kernel features might pose challenges for practitioners. Additionally, a broader comparative analysis and more detailed examination of the algorithm’s performance across different distance metrics would strengthen the paper’s contributions.

**Strengths:**

1. sDBSCAN significantly improves the scalability of density-based clustering algorithms, enabling the handling of million-point datasets efficiently.
2. By utilizing cosine distance and random projections, sDBSCAN effectively handles high-dimensional data.
3. The extension of sDBSCAN to other distance metrics (L2, L1, χ², and Jensen-Shannon) via random kernel features enhances its applicability across different types of data.

**Weaknesses:**

1. The algorithm’s performance may depend on the selection of parameters, and the sensitivity to these parameters is not deeply explored.
2. The use of random projections and kernel features might complicate the implementation and understanding of the algorithm for practitioners.
3. While the algorithm is extended to other distance metrics, the performance and effectiveness of these extensions are not thoroughly evaluated.
4. The paper could benefit from a more comprehensive comparative analysis with a broader range of clustering algorithms beyond DBSCAN variants.

**Questions:**

1. How sensitive is sDBSCAN to the choice of parameters such as the number of random vectors and the radius for core points? Are there guidelines for setting these parameters?
2. How does sDBSCAN perform when extended to L2, L1, χ², and Jensen-Shannon distances in terms of both accuracy and speed?
3. How robust is sDBSCAN to variations in data distribution and the presence of noise and outliers?
4. Are there specific real-world applications or case studies where sDBSCAN has shown significant advantages over other clustering methods?

---

> ### Author Rebuttal · Authors · 2024-08-04
>
> Thanks for your reviews. However, we found that the raised weaknesses and questions have been addressed in the paper and Appendix (Section B).
> We explain in detail below.
>
> **W1) The algorithm’s performance may depend on the selection of parameters, and the sensitivity to these parameters is not deeply explored.
> Q1) How sensitive is sDBSCAN to the choice of parameters such as the number of random vectors and the radius for core points? Are there guidelines for setting these parameters?**
>
> One of our contributions is sOPTICS, a visual tool to guide the setting of ε, radius for core points, given minPts.
>
> For ε, we use sOPTICS graphs to select the relevant range, then find 6 values on such range to run sDBSCAN.
> Fig 1 shows the sOPTICS graphs for L1, L2, Cosine, JS, and sDBSCAN's accuracy on 6 values of ε from a certain range that separates the valleys on sOPTICS graphs.
> E.g. on Cosine (Fig 1c and 1g), we select ε = {0.1, 0.11, 0.12, 0.13, 0.14, 0.15}.
> We detail it in Section 4.1, especially in the 2nd and 3rd paragraphs.
> In Section 4.2 (Pamap2 and Mnist8m), we state "We use sOPTICS graph (see the supplement) to select relevant ranges of ε".
> These graphs are Fig 5 and 6 in Appendix.
> This is how we select ε for Fig 2-3.
>
> For minPts, Fig 1-3 use minPts = 50.
> In Section B.8, Fig 13, 15 show sOPTICS graphs with minPts = 100.
> Again, we set ε based on the suggested range from sOPTICS graphs to get sDBSCAN's accuracy in Fig 14, 16.
>
>
> sDBSCAN requires extra parameters, including number of random vectors D, top-k closest/furthest random vectors, top-m points closest/furthest to random vectors.
> For L1, L2, χ², JS distances, sDBSCAN need the number of embeddings d', and the scale σ.
> We have stated on Page 8 (Parameter settings) that "Experiments on the sensitivity of parameters m, k, σ, d', minPts of sDBSCAN and sOPTICS are left in the supplement."
>
> - For D, we need D = power of 2 to use FHT (see Section 3.3 - Reduce random projection cost).
> On Mnist, Mnist8m d = 784, we set D = 1024.
> D = 512 works well on Pamap2 with d = 51.
> For consistency, we use D = 1024 throughout the paper.
>
> - For sensititvity of m, k, Section B.2 shows how to select sDBSCAN's parameter setting and its running time.
> Section B.7 shows the sensitivity of m, k on L1 on Pamap2 (in Fig 12).
>
> - For sensitivity of σ for L1, L2, see Section B.6.
> Fig 7 for sDBSCAN and Fig 8-9 for sOPTICS with L1, L2 on Pamap2.
>
> - For sensititvity of d', see Fig 10-11 for sDBSCAN and sOPTICS on χ², JS on Mnist.
>
> We do not tune these parameters carefully, except ε guided by sOPTICS. The same setting D = 1024, m = minPts, σ = 2ε, d' = D, k = {5, 10} works well on all experiments (see Parameter settings in Section 4 and Tab 2 for data set property).
>
> **W2) The use of random projections and kernel features might complicate the implementation and understanding of the algorithm for practitioners.**
>
> The random projections and kernel features are building blocks of many practical machine learning algorithms.
> Scikit-learn has featured many randomized kernel features, and we use two of them [19, 20].
> [19] won The Test-of-Time Award while [20] is popular in computer vision.
> Section A.3 shows how to guarantee distance preservation given d' = O(log(n) kernel features.
>
> The random projection is the algorithmic primitive for many machine learning problems, including clustering [a, b]
>
> [a] NIPS 10 - Random Projections for k-means Clustering
>
> [b] ICML 03 - Random Projection for High Dimensional Data Clustering: A Cluster Ensemble Approach
>
> **W3) While the algorithm is extended to other distance metrics, the performance and effectiveness of these extensions are not thoroughly evaluated.
> Q2) How does sDBSCAN perform when extended to L2, L1, χ², and JS distances in terms of both accuracy and speed?**
>
> The accuracy of sDBSCAN on L2, L1, χ², JS were shown in **all** figures in the paper and Appendix.
> The speed is shown in the caption.
> Tab 5-7 for Cosine, L1 on Pamap2, and for Cosine, L2, χ², JS on Mnist8m.
> Fig 7-16 for parameters sensitivity on L1, L2, χ², JS.
>
> **Q3) How robust is sDBSCAN to variations in data distribution and the presence of noise and outliers?**
>
> Similar to DBSCAN, sDBSCAN labels all outliers as the noise class.
> The comparison between sDBSCAN and exact DBSCAN on Mnist and Pamap2 (Fig 1-2, Tab 5), showing the robustness of sDBSCAN with presence of noise.
> Tab 3 in Section B.2 shows sDBSCAN can recover nearly the DBSCAN's output with 95% NMI.
>
> For various data distributions, our experiment data has various # clusters (Tab 2), and hence the data set's distribution varies significantly.
> Also, the valleys of sOPTICS graphs have very different shapes.
>
> The sDBSCAN-1NN (Section 3.3) partially addresses the variations in data distribution.
> If minPts is set correctly, we can identify many core points in the high-density regions and a few core points in low-density regions.
> Since sDBSCAN-1NN uses the nearest found core points to assign cluster labels, it can link points in the low-density regions together, and points in high-density regions together.
> Fig 3, 16: sDBSCAN-1NN shows higher accuracy compared to sDBSCAN.
>
> **W4) The paper could benefit from a more comprehensive comparative analysis with a broader range of clustering algorithms beyond DBSCAN variants.
> Q4) Are there specific real-world applications or case studies where sDBSCAN has shown significant advantages over other clustering methods?**
>
> Tab 5: Compared to k-means++, sDBSCAN is faster and 10% NMI higher on L1.
> Fig 3, 16: compared to kernel k-means, sDBSCAN gives similar NMI and runs on a single machine.
> In Section B - Clustering competitors: We state "We tried several clustering algorithms on scikit-learn, including spectral clustering, kernel k-means. They could not work on million-point data sets given our DRAM of 128GB."
>
> sDBSCAN is non-parametric but still competitive with k-means variants that need the value of k (unknown in practice) regarding both accuracy and running time.

---

> > ### Comment · Reviewer_KGU7 · 2024-08-09
> >
> > The authors addressed some of my concerns in this rebuttal. I would like to increase my score to 4 instead of 3.

---

> > > ### Author Response · Authors · 2024-08-09
> > >
> > > Thanks for your comments and your score increase.
> > >
> > > If possible, could you let us know the concerns that you are not satisfied and the rationale, so that we have chance to clarify and change your opinion on our work?

---

> > > > ### Comment · Reviewer_KGU7 · 2024-08-14
> > > >
> > > > Dear authors, I have read your rebuttal and my evaluation remains unchanged

---

> > > > > ### Author Response · Authors · 2024-08-14
> > > > >
> > > > > Thanks for your comment. Since your confidence is 5 and you have not indicated which of your concerns have not been addressed and its rationale, we feel that we have not had any chance to clarify further.

---

### Official Review · Reviewer_gFWb · 2024-07-08

**Soundness:** 3
**Presentation:** 2
**Contribution:** 2
**Rating:** 5
**Confidence:** 4

**Summary:**

This paper proposes a scalable DBSCAN algorithm which facilitates random projection to quickly approximate the $\varepsilon$-neighborhood. The proposed algorithm speeds up conventional DBSCAN algorithms by orders of magnitude.

**Strengths:**

S1. The proposed algorithm speeds up conventional DBSCAN algorithms by orders of magnitude.

S2. The formal proof as well as time complexity analysis are presented in the paper.

S3. The authors also provide a multi-thread implementation of the proposed algorithm for much faster execution.

**Weaknesses:**

W1. The main contribution may not be very high, because the core idea of approximating $\varepsilon$-neighborhood is borrowed from the previous work [22].

W2. It is very important to provide a heuristic or guideline for setting the value of $\varepsilon$, because the peak reached at different values among the proposed algorithm and previous algorithms. For this purpose, the authors present a variation of OPTICS, i.e., sOPTICS, which suggests a **range** of $\varepsilon$. As this range is quite wide, I am skeptical about the practical usability of the proposed algorithm. How do you pick the best value while you do not know the ground-truth clustering result in practice?

W3. More datasets need to be included for the evaluation. Only three datasets may not be sufficient to show the superiority of the proposed algorithm.

W4. The organization of this paper has a critical problem. Even though it is claimed that there are two important algorithms, sDBSCAN and sOPTICS, the entire description of sOPTICS is only presented in the supplementary material.

W5. It is not clear whether the preprocessing time is included in the reported execution time. If not, I believe that the proposed algorithm is too much favored in the evaluation.

W6. The clustering accuracy is often lower than those of the conventional algorithms at the cost of fast execution. To achieve a good accuracy, more hyperparameters need to be tuned carefully.

W7. (Minor) NeurIPS may not be the best venue for this paper. KDD best suits this paper.

---

After rebuttal, I have increased my rating to 5.

**Questions:**

See W1~W6.

**Limitations:**

The authors discussed the limitations about a sufficiently large number of random vectors, which sound reasonable.

---

> ### Author Rebuttal · Authors · 2024-08-04
>
> Thanks for your reviews. We will address the raised weakness below.
>
> **W1. The main contribution may not be very high, because the core idea of approximating ε-neighborhood is borrowed from the previous work [22].**
>
> sDBSCAN uses the recent result in [22], i.e. the property of random projections given the number of vectors D is significantly large.
> However, this new property has **not** been explored in any clustering algorithm.
> Random projection is used as dimensionality reduction with D = O(log(n)) to speed up the k-means [a] and DBSCAN [13].
> We found that [13] does not offer theoretical guarantees on the DBSCAN result, and we could not even run [13] on Mnist with n=70,000.
>
> Building on the theory of [22], we prove that sDBSCAN can recover the DBSCAN's output with good probability.
> This theoretical result aligns with previous works, e.g. [a] that guarantees the quality of k-means using random projections.
> For practicality, we show that by considering only top-minPts points closet to the random vectors, sDBSCAN can detect core points with high probability (Lemma 3).
> This scales up sDBSCAN in both time and **space** complexity.
>
> We found that space complexity is the main issue of several kernel-based clustering and DBSCAN since we have to maintain O(n^2) pairwise distances.
> Our machine with 128GB of RAM cannot run spectral clustering or DBSCAN/OPTICS provided by scikit-learn with million-point data sets.
> sDBSCAN/sOPTICS can run on Mnist8m of size 42GB due to their low memory overheads.
>
> Therefore, we believe that our contribution (especially in practice) is significant.
> We hope sDBSCAN and its parameter guideline sOPTICS that support various distances (Cosine, L1, L2, χ², JS) will be as popular as k-means in clustering analysis.
>
> [a] NIPS 10 - Random Projections for k-means Clustering
>
> **W2. The authors present a variation of OPTICS, i.e., sOPTICS, which suggests a range of ε. As this range is quite wide, I am skeptical about the practical usability of the proposed algorithm. How do you pick the best value while you do not know the ground-truth clustering result in practice?**
>
> Selecting ε without ground truth is exactly the contribution of sOPTICS graphs where each valley corresponds to a cluster.
> We can also select better metrics to run sDBSCAN based on sOPTICS graphs with clear valleys (discussed in Sec 4.1).
>
> On Cosine (Fig 1c), we select ε in [0.1 0.15] as any ε in this range can separate the valleys of the sOPTICS graph.
> Among 6 values of ε {0.1, 0.11, 0.12, 0.13, 0.14, 0.15}, sDBSCAN (Fig 1g) can reach DBSCAN's accuracy.
> On the selected range based on sOPTICS, one value of ε gives the accuracy peak on all 3 data sets even with various parameter settings (minPts, σ, d' on Section B6, B7, B8).
>
> The sampling competitors: sngDBSCAN [15], DBSCAN++ [14] do not offer any tool to suggest ε, and different sampling strategies even require different optimal values of ε!
>
> As sDBSCAN can nearly recover DBSCAN's output with NMI of 95% (Tab 3 in Appendix) with the same value of (ε, minPts), if DBSCAN works well on a data set without ground truth, sDBSCAN can achieve similar performance but run much faster.
>
> **W3. More datasets need to be included for the evaluation. Only three datasets may not be sufficient to show the
> superiority of the proposed algorithm.**
>
> We agree and will add new datasets (e.g. KDDCup or deep learning-based pretrained data sets).
>
> **W4. The organization of this paper has a critical problem. Even though it is claimed that there are two important
> algorithms, sDBSCAN and sOPTICS, the entire description of sOPTICS is only presented in the supplementary material.**
>
> We could not detail sOPTICS in the main paper given 9 page limits.
> Similar to OPTICS, sOPTICS is a visual tool to guide the setting of ε.
> OPTICS requires many new concepts, including core distance, reachability distance, and its algorithm requires significant room to present (see Section A1).
> We decided to move sOPTICS to Appendix as its algorithm description is rather simple given OPTICS's description.
> While sOPTICS is also one of our contributions, we have to focus on sDBSCAN on the main paper due to the limited space.
>
> **W5. It is not clear whether the preprocessing time is included in the reported execution time. If not, I believe that the
> proposed algorithm is too much favored in the evaluation.**
>
> The reported execution time **includes** the preprocessing time.
> Tab 4 in Appendix shows the running time of preprocessing, finding core points, and clustering.
> The execution of preprocessing (with sequential memory assess) is rather small, just around 10%, and is dominated by the distance computations (with random memory assess) for finding core points (See **Time complexity** in Section 3.4)
>
> **W6. The clustering accuracy is often lower than those of the conventional algorithms at the cost of fast execution. To
> achieve a good accuracy, more hyperparameters need to be tuned carefully.**
>
> It is not totally correct.
>
> Tab 5: Compared to k-means++, sDBSCAN is faster and gives 10% NMI higher on L1.
> Fig 3, 16: Compared to kernel k-means, sDBSCAN gives similar NMI (only 1-2% off) but runs on a single machine.
> Note that we use k as the ground-truth number of clusters.
>
> sDBSCAN requires extra parameters, including number of random vectors D, top-k closest/furthest random vectors, top-m points closest/furthest to random vectors. For L1, L2, χ², JS distances, sDBSCAN need the number of embeddings d', and the scale σ.
> Section B6-B8 shows the sensitivity of these parameters.
> We also explained the details in the rebuttal message to Reviewer #KGU7
>
> We do not tune these parameters carefully, except ε visually guided by sOPTICS.
> The same setting D = 1024, m = minPts, σ = 2ε, d' = D, k = {5, 10} works well on all experiments (see **Parameter settings** in Section 4 and Tab 2 for data set property).
> Without ground truth, we have to run k-means with several k and DBSCAN variants with several ε on a large range (e.g. on L1, L2).

---

> > ### Comment · Reviewer_gFWb · 2024-08-09
> > **Thank you for your responses**
> >
> > Thank you for your responses. I have carefully read the authors' rebuttal.
> >
> > The responses for W1, W2, and W5 are understandable, so I would like to increase my rating to 5. For W6, I meant Figure 2 and Figure 3, where the peaks of sDBSCAN are often lower than those of the other algorithms.

---

> ### Author Response · Authors · 2024-08-09
>
> Thanks for clarification and your score increase. We would like to explain further for W4 and W6
>
> **W4. The organization of this paper has a critical problem. Even though it is claimed that there are two important algorithms, sDBSCAN and sOPTICS, the entire description of sOPTICS is only presented in the supplementary material.**
>
> The camera-ready has one extra page. If the submission get accepted, we will use this extra page to explain OPTICS and sOPTICS. This would make the paper stand-alone and clearify the contributions of our work.
> We feel it is hard to squeeze both sOPTICS and sDBSCAN into the current limits of 9 pages.
> We hope you understand.
>
> **W6. The clustering accuracy is often lower than those of the conventional algorithms at the cost of fast execution. To achieve a good accuracy, more hyperparameters need to be tuned carefully.**
>
> We agree that sDBSCAN's peaks are lower than the other algorithms in Fig 2 and 3, and would like to add further explanation.
>
> Compared to sngDBSCAN, sDBSCAN's peak is often higher on all studied metrics on both Fig 2 and 3.
>
> In Fig 2, sDBSCAN's peaks are lower than uDBSCAN++ on L2, and lower than both uDBSCAN++ and kDBSCAN++ on Cosine.
> However, sDBSCAN's on L1 is higher than those peak values on L2 and Cosine.
> On Tab 5 (Appendix), sDBSCAN on L1 reach 46% NMI, while uDBSCAN++ reaches 46% and kDBSCAN++ reaches 39% on cosine.
> This indeed shows the advantages of sDBSCAN that can work on several metric distances, while DBSCAN++ works only on L2 (given the current theory and released implementation).
>
>
> We also note that the range of $\epsilon$ is found by sOPTICS.
> Without such recommendation, the cost of finding relevant $\epsilon$ for sampling DBSCAN variants can be as inefficient as running an exact OPTICS (i.e. $O(n^2)$ in both space and time).
> To add an evidence, sngDBSCAN [15] studied several low-dimensional million-point data sets. Their Fig 4 shows that good $\epsilon$ for each data set is on very different ranges $[0.5, 2.0], [0, 12], [0, 0.04], [1, 5], [0, 0.08]$ !!!
>
> In Fig 3, sDBSCAN could not reach the accuracy of kernel k-mean. This experiment setting is less favoured to sDBSCAN.
> While kernel k-means use the prior knowledge k = 10 and run on a supercomputer, sDBSCAN selects $\epsilon$ based on sOPTICS without any ground-truth, and run on a single machine.
>
> _________
>
> Thank again for your comment, and please let us know any other concerns that we should address to change your opinion on our work.

---

> > ### Comment · Reviewer_gFWb · 2024-08-10
> > **Thank you for your clarification**
> >
> > Thank you for your further clarification. **In practice**, it is hard to pick the best distance measure, since we do not know the NMIs in real-world applications. Also, the original DBSCAN algorithm is not restricted to a specific distance measure (I am not familiar with DBSCAN++). Overall, I would like to stick to the current rating.

---

> > > ### Author Response · Authors · 2024-08-10
> > >
> > > Thanks for your comment.

---

### Official Review · Reviewer_UCFw · 2024-07-08

**Soundness:** 2
**Presentation:** 2
**Contribution:** 3
**Rating:** 5
**Confidence:** 3

**Summary:**

The authors present an accelerated variant of DBSCAN based on random projections. Theoretical results are provided that indicate that this method will yield a similar clustering as the original DBSCAN. Experiments on real-world data show that this method indeed achieves similar performance at a fraction of the computational cost.

**Strengths:**

The proposed method seems to significantly improve upon the running time of DBSCAN while yielding similar performance.

The authors support their claims with theoretical proofs.

**Weaknesses:**

The paper is hard to read due to many language errors, odd sentences and sloppy formulations.

Lemma 1 seems to be a crucial result for this work, but the proof is not given. Even if it was proven in prior work, it might be worth-while to include the proof. At the very least, the formulation of Lemma 1 should be greatly improved, because the current formulation is rather sloppy:
* $\mathcal{S}^{d-1}$ in Lemma 1 is not introduced.
* The $\sim$ in Lemma 1 is usually used to indicate that the LHS follows the distribution given on the RHS. But here, the result seems to concern some convergence in distribution, which is usually denoted by $\stackrel{\mathcal{D}}{\rightarrow}$. Please improve the formulation of this important Lemma.
* When you say that the coordinates of $r_i$ are drawn from a standard Gaussian, then you cannot simply say "assume w.l.o.g." that $r_1=\arg\max_i|q^\top r_i|$. What you could do instead, is formulate your result in terms of $r^*=\arg\max_i|q^\top r_i|$.

The method is validated using the NMI. NMI is known the be biased towards fine-grained partitions [1,2]. Please use a different validation measure, like the Adjusted Mutual Information [1] or the correlation coefficient [2]

Minor comments:
* Line numbers would be helpful in the reviewing process. Please use the NeurIPS latex template without changing any of the settings.
* The plots shown in Figure 1 are too small and the labels are not readable.

[1] Vinh, N. X., Epps, J., and Bailey, J. Information theoretic measures for clusterings comparison: Variants, properties,
normalization and correction for chance. The Journal of Machine Learning Research, 11:2837–2854, 2010.
[2] Gösgens, M. M., Tikhonov, A., & Prokhorenkova, L. (2021, July). Systematic analysis of cluster similarity indices: How to validate validation measures. In International Conference on Machine Learning (pp. 3799-3808). PMLR.

**Questions:**

How fast should $D$ grow be in terms of $d,n$ in order for Lemma 1 to hold?

The assumption regarding $t$ described between Lemma 2 and Theorem 1 (once again: please enable line numbers) seems somewhat strong. Could you comment on this?

In Lemma 3, $D=n^{1/k\alpha^2_*}$ is written. Does this mean $D=n^{\alpha^2_*/k}$ or $D=n^{k^{-1}\alpha_*^{-2}}$? Please change this expression for clarification.

**Limitations:**

Usage of NMI and missing proof of Lemma 1 (see above)

---

> ### Author Rebuttal · Authors · 2024-08-06
>
> Thanks for your reviews. We will address the raised weaknesses and questions below.
>
> **W1: Regarding notation and proof of Lemma 1**
>
> We agree with your comments regarding sloppy formulation and will fix them all.
> Since we start sDBSCAN with cosine distance, we state Lemma 1 as a simplified version of [22] on unit sphere $S^{d-1}$
> Indeed, Lemma 1 is similar to Lemma 2 in [23] as [23] uses this result to scale up approximate kNN on cosine.
>
> We will add a proof of Lemma 1 in the appendix to make the paper stand-alone.
> Lemma 1 follows the results of [22] in case data points are on a unit sphere, and it holds when $D \rightarrow \infty$.
>
> **W2: The validation measure NMI, Adjusted Mutual Information (AMI) or the correlation coefficient**
>
> We have both AMI and NMI. Both scores are very similar (within 1 ‰) in our empirical results.
> The reason might be that the 3 used data sets do not have any small clusters.
> We report NMI since we want to compare it with the reported NMI of the kernel k-means [21] as [21] does not have AMI scores on Mnist8m.
> We found that sngDBSCAN's AMI is even lower than its NMI (See the figure in response.pdf).
>
> If we understand correctly, to compute the correlation coefficient, we have to form the pair-counting indices vector of size $n^2$ where each component (i, j) = 1 if $x_i$ and $x_j$ are in the same cluster.
> Such measurement seems nontrivial to compute given $n$ > 1M points (Pamap2 and Mnist8m) due to its space complexity.
>
> **Q1) How fast D grows with d, n to ensure Lemma 1 holds?**
>
> For our theoretical analysis, we use Lemma 1 with $D \rightarrow \infty$, and we stated on Limitations.
>
> The non-asymptotic results of Lemma 1 are complicated.
> We will sketch it here, and will add a full proof into the appendix.
>
> Let $x, q \in S^{d - 1}$ and $D$ random vectors $r_i$ whose coordinates are randomly selected from $N(0, 1)$.
> Given $\rho = x^{\top}q$, and let $X_i = x^{\top}r_i, Q_i = q^{\top}r_i$, we  have $(X_i, Q_i) \sim N(0, 0, 1, 1, \rho)$.
> In particular, $X_i = Q_i * \rho + Z_i$ where $Z_i \sim N(0, 1 - \rho^2)$
>
> We assume that $r_* = \arg\max_{r_i} | q^{\top}r_i | = \arg\max_{r_i}  q^{\top}r_i $.
> $Q_*$ is the maximum value of $D$ random normal variables.
>
> We first show that $D = e^{1/\delta^2}$ for $0 < \delta = 1 / \sqrt{\log{D}}< 1$, then with probability at least $1 - 2\delta$,
> $\sqrt{2 \log{D}} (1 - o(\log{\log{D}} / \log{D})) \leq Q_* \leq \sqrt{2 \log{D}}$.
>
> After that, we use Chernoff tail bounds of normal variables $Z_* = X_* - \rho * Q_*$ to bound the probability that $X_*$ deviates from its expectation, i.e. $\rho \sqrt{2 \log{D}}$.
>
> While the Gaussian distribution of $X_*$ in Lemma 1 requires $D \rightarrow \infty$, the non-asymptotic tail bounds of $X_*$ show the concentration of $X_*$ around its expectation with high probability $1 - 2 / \sqrt{\log{D}}$.
>
> On Lemma 3, we show that if $D = n^{1 / (k \alpha^2)}$ where $\alpha$ depends on the data distribution, and assume that such $D$ is large enough to ensure Lemma 1 holds, then we can find core points by maintaining just $minPts$ points closest to each random vector.
> This saves the space and time complexity of sDBSCAN in practice.
> We found that $D = 1024$ works well in 3 used data sets, including Mnist8m with n = 8.1M, d = 784.
> We think it is due to the small world phenomenon "Neighbors of neighbors tend to be neighbors".
> If $x$ is closest to the random vector $r_*$, then $x$ tends to be close to $r_*$'s neighbors.
>
> **Q2) The assumption regarding the strong connection of a density-based cluster $C_i$ of size $n_i$: For any two close core points $x, y \in C_i, dist(x, y) < \epsilon$, their neighborhoods have to share at least $t = O(\log{n_i})$ common core points.
> With this assumption, sDBSCAN can recover DBSCAN cluster $C_i$ with probability at least $1 - 1/n_i$.**
>
> We note that density-based clustering will link only core points together to form the cluster skeleton.
> Hence this assumption is needed to guarantee any density-based clustering.
> Consider the case where core points $x, y, dist(x, y) = \epsilon$ where $B(x, \epsilon) \cap B(y, \epsilon) =$ {$x, y$}.
> There is only one edge $xy$ connecting $B(x, \epsilon)$  and $B(y, \epsilon) $.
> In the worst case, if any approximate DBSCAN misses identifying $xy$ while approximating $B(x, \epsilon)$ and $B(y, \epsilon)$, it cannot recover DBSCAN's output.
> The assumption $t = O(\log{n})$ makes the density-based cluster $C$ not thin at anywhere, i.e. there are $t$ paths $x p_i y$ connecting $x, y$ together where the core point $p_i \in B(x, \epsilon) \cap B(y, \epsilon)$.
>
> When $minPts = 50$ (note that $t$ = log(8M) = 16), we expect many core points will have more than 16 core points within their $\epsilon$-neighborhoods.
> This happens in practice since if the core point $x$ tends to be in high-density regions and $|B(x, \epsilon)| >> minPts$.
> Since we use the minPts points closest to random vectors to find $\epsilon$-neighborhoods of core points, if both core points share the same closest random vectors, their neighborhoods tend to contain the same core points, and hence be connected together.
>
> If $t$ is not large enough, there might be a thin region separating the density-based cluster.
> Then we should choose $\epsilon$ larger to increase the number of points in $\epsilon$-neighborhood, and hence increasing $t$.
> In practice, we use sOPTICS graphs to visualize the cluster structure and to select $\epsilon$ large enough to separate valleys.
> This will give good accuracy.
> Fig 1c, 1d shows that $\epsilon$ should be closer to 0.15 to separate 4 clusters, and Fig 1g, 1h shows that $\epsilon$ = 0.14 reaches the peak.
>
> A variant of the assumption of $t$ was used in [15] to guarantee the recovery of sngDBSCAN on the DBSCAN's output.
> Besides, [15] also needs other 2 strong assumptions regarding the data distribution (See Assumption 1 in [15]).
>
> **Q3) Regarding D in Lemma 3.**
>
> We will fix it: $D = n^{k^{-1} \alpha_*^{-2}}$

---

> ### Comment · Reviewer_UCFw · 2024-08-12
>
> I thank the authors for their rebuttal.
>
> >  Lemma 1 follows the results of [22] in case data points are on a unit sphere, and it holds when $D \rightarrow \infty$.
>
> This does not answer my question concerning what limit is proven. Does Lemma 1 concern a limit in distribution (weak limit) or an exact result (the r.v. exactly follows the normal distribution) for $D$ above some threshold?
>
> > we have to form the pair-counting indices vector of size $n^2$ where each component (i, j) = 1 if $x_i$ and $x_j$ are in the same cluster.
>
> This is incorrect. One does not have to compute these pair-counting vectors in order to compute the value of this measure. This measure (and any other pair-counting measure) can be computed in $O(n)$, as explained in [2]. Actually, computing the correlation is faster than computing AMI, which may have quadratic complexity if one of the partitions consist of many clusters.
>
> > We found that sngDBSCAN's AMI is even lower than its NMI (See the figure in response.pdf).
>
> One cannot compare AMI values to NMI values. They are different measures.  One can, however, compare how the two measures rank a given set of clustering methods. In general, reporting AMI values is preferred over reporting NMI values, because NMI is a biased measure. I can understand that you might want to use NMI to compare your results to previous work, but please report AMI (or correlation) in all other cases.
>
> Moreover, I don't understand the plots in rebuttal.pdf: your x-axes have NMI and AMI written on them, but with values exceeding 1000. Both NMI and AMI are upper-bounded by 1 (or by 100, if expressed as 'percentages').
>
> > For our theoretical analysis, we use Lemma 1 with $D \rightarrow \infty$, and we stated on Limitations.
>
> $D\rightarrow\infty$ as $n\rightarrow\infty$? This would allow for $D=\log\log n$. Are you sure this is sufficient?
>
> The proposed method definitely is certainly a great contribution, but the theoretical validation is unacceptable. In particular, the asymptotics seems quite sloppy. I will decrease the rating.

---

> ### Author Response · Authors · 2024-08-13
>
> Thank for your comments and we are sorry that our rebuttal does not satisfy your request. We address your main concerns below.
>
> **New measurements (CC) instead of NMI / AMI**
>
> We went through the paper (indeed its arxiv version) [2] you suggested and found the way to compute the correlated coefficient (CC) between two clustering labels in $O(n)$ times. Using sklearn.metrics.cluster.pair_confusion_matrix() to compute N11, N10, N01, N00 values, we can compute any kinds of pair-counting based scores on Table 5 (Appendix B), including CC. We have generated a new figures on NMI, AMI, CC on Mnist and Pamap, but cannot update the response.pdf. So we add a table regarding CC on Mnist with cosine here as a reference.
>
> | $\epsilon$ | sDBSCAN | sngDBSCAN | uDBSCAN | kDBSCAN | DBSCAN |
> |----------------|----------------|-------------------|----------------|----------------|--------------|
> | 0.1 | 0.13 | 0.072 | 0.039 | 0 | 0.142 |
> | 0.11 | 0.13 | 0.083 | 0.07 | 0 | 0.169 |
> | 0.12 | 0.134 | 0.09 | 0.093 | 0 | 0.164 |
> | 0.13 | 0.145 | 0.097 | 0.094 | 0 | 0.144 |
> | 0.14 | 0.166 | 0.104 | 0.096 | 0 | 0.132 |
>
> Regarding the CC measure, sDBSCAN is still better than the other sampling DBSCAN variants on the suggested range of $\epsilon$.
> ________________________________________________________
> For the reponse.pdf, the x-axis is the value of $\epsilon$ suggested by the sOPTICS graphs.
> The y-axis is the NMI and AMI scores in percentages, measured at 6 different values of $\epsilon$.
> We observe that the NMI and AMI scores are very similar, e.g. Fig 1(a) at $\epsilon = 0.14$, sDBSCAN has NMI = 0.4139, AMI = 0.4132; uDBSCAN has NMI = 0.3112, AMI = 0.3095, but sngDBSCAN has NMI = 0.3170, AMI = 0.2678.
> Similar to Pamap data set, the difference between AMI and NMI scores of all methods is negligible (within 0.002).
>
> **The asymptotic theoretical analysis of Lemma 1**
>
> Lemma 1 holds given the limit in distribution (weak limit).
> That is, given $r_*$ is closest to $q$ among $D$ random vectors, then $x^T r_* \xrightarrow{D} N(x^T q \sqrt{2 \log{D}}, 1 - (x^T q)^2)$.
>
> This is the main finding of [22], which shows the connection between random projections when $D \rightarrow \infty$ and the asymptotic behavior of the concomitants of extreme order statistics [18].
> It has been used to speed up nearest neighorbor search [23].
>
> [22] Simple Yet Efficient Algorithms for Maximum Inner Product Search via Extreme Order Statistics - KDD 21
>
> [23] Falconn++: A Locality-sensitive Filtering Approach for Approximate Nearest Neighbor Search - NeurIPS 22
>
> [18] The Asymptotic Theory of Concomitants of Order Statistics - Journal of Applied Probability 74
> ________________________________________________________
>
> Regarding your question about non-asymptotic results, as far as we know, it seems impossible to achieve the normal distribution given any relationship between $D$ and $n$.
> We have not found any result that studies the non-asymptotic behavior of extreme order statistics and its concomitants.
> We conjecture that such (useful) non-asymptotic results might not exist due to the field of extreme value theory.
>
> Our previous rebuttal shows our effort to achieve non-asymptotic results.
> We first bound the value of $Q_*$, the maximum value of $D$ independent normal variables.
> Another approach using Fisher–Tippett–Gnedenko theorem to estimate $E[Q_*]$ can be seen here (https://en.wikipedia.org/wiki/Generalized_extreme_value_distribution).
>
> Set $\delta = 1 / \sqrt{\log{D}}$, we have $\sqrt{2 \log{D}} (1 - o(\log{\log{D}} / \log{D}) < Q_* < \sqrt{2 \log{D}}$ with prob. $1 - 2\delta$.
> To ensure $Q_*$ is highly concentrated on its expectation, i.e. $E[Q_*] \sim \sqrt{2 \log{D}}$, we need $D = e^{n^4}$.
> The questioned setting $D = \log{\log{n}}$ does not work as $\log{\log{D}} / \log{D}$ is not small enough.
>
> When $D = e^{n^4}$, we have $\delta = 1/n^2$.
> By the union bound, we can show that **all** $n$ random variables $Q^i_*$ corresponding to the point $x_i$ will be around its expectation with probability $1 - 1/n$.
> After that, using the bivariate normal distribution between $X_i = x^T r_i$ and $Q_i = q^T r_i$, we can bound the tail of $x^T r_*$ where $r_*$ is the closest random vector to $q$.
>
> The setting of $D = e^{n^4}$ to ensure the concentration of maximum of $D$ normal variables around $\sqrt{2 \log{D}}$ is pessimistic since it applies to the worst-case distribution of data.
> We found that there are several papers (in theory) that use $Q_* = \sqrt{2 \log{D}}$ to derive their results, for example
>
> - Lossy Compression via Sparse Linear Regression: Computationally Efficient Encoding and Decoding. IEEE Trans. Inf. Theory 60(6): 3265-3278 (2014)
> - Distributed Estimation of Gaussian Correlations. IEEE Trans. Inf. Theory 65(9): 5323-5338 (2019)
>
> If we use $Q_* = \sqrt{2 \log{D}}$ as theoretical results above, Lemma 1 holds as $x^T r_* = \rho \sqrt{2 \log{D}} + Z_i \sim N( \rho \sqrt{2 \log{D}}, 1 - \rho^2)$ where $\rho = x^T q$ and $Z_i \sim N(0, 1 - \rho^2)$.

---

> > ### Comment · Reviewer_UCFw · 2024-08-13
> >
> > > New measurements (CC) instead of NMI / AMI
> >
> > Thank you for your updated results. It is indeed clear that sDBSCAN outperforms the other sampling methods.
> >
> > > Lemma 1 holds given the limit in distribution (weak limit).
> >
> > Thank you for clarifying. It is important to update this in the paper, because the '$\sim$' notation in a limit $D\rightarrow\infty$ is simply not sound.
> >
> > > That is, given $r_*$ is closest to $q$ among $D$ random vectors
> >
> > I really think you should formulate the lemma by *defining* $r_*$ to be the closest to $q$ among $D$ random vectors, because this is cleaner in a probabilistic setting than using language like "assuming" or "given".
> >
> > > Regarding your question about non-asymptotic results
> >
> > I did not ask about any non-asymptotic results. Previously, I was under the impression that we also needed $n\rightarrow\infty$ for Lemma 1 to hold, but I now see that this is not the case. In such double limits, it is always worth asking how fast/slow one quantity can grow with respect to the other, which is why I asked about $D=\log\log n$. But I now see that this question does not make any sense.
> >
> > I am willing to increase my rating. But I do this hoping that the authors will improve the formulation of Lemma 1 (describing the weak limit, defining $r_*$ adequately and adding a proof in the appendix) and improve the experimental validation (report AMI / correlation instead of NMI, except possibly when comparing to other papers). This paper is a solid contribution and sDBSCAN is clearly a great improvement upon other sampling methods. You have the theoretical and experimental results to show this, but their presentation just needs some improvement.

---

> > > ### Author Response · Authors · 2024-08-13
> > >
> > > Thanks for your comment and update the score.
> > >
> > > We will fix all sloppy notations of Lemma 1 as your suggestions.
> > > We will report both AMI and CC for Mnist, Pamap, and Mnist8m data sets.
> > > We also keep NMI on Mnist8m for kernel k-means on the appendix, and cite the two papers you suggested regarding the cluster validation.

---

### Official Review · Reviewer_C5J7 · 2024-07-08

**Soundness:** 3
**Presentation:** 3
**Contribution:** 3
**Rating:** 7
**Confidence:** 2

**Summary:**

DBSCAN is a popular density-based clustering algorithm. For a parameter $\epsilon$, a point $p \in X$ is core if its $\epsilon$-ball is large (over $minPts$ in size). Core points within each others' $\epsilon$-balls are then connected via an edge, non-core points within $\epsilon$-balls are considered cluster borders, and all other non-core points are considered noise. OPTICS is an algorithm which visualizes, across all $\epsilon$ selections, the density of a clustering to assist in the selection of $\epsilon$.

This paper focuses on improving the space efficiency of the first step of DBSCAN, which finds the $\epsilon$-neighborhoods of all points. Normally this requires $O(n^2)$ space. Following sampling-based approaches, they first construct a random projection-based index which allows them to identify $\epsilon$-neighborhoods without fully constructing them. This pre-existing method projects points onto vectors whose indices are $~N(0,1)$, and leverages the fact that if $r$ is the random vector maximizing $|q^Tr|$, then $|x^Tr| \approx |x^Tq|$. Thus, $\epsilon$-balls can be approximated using the top-$k$ random vectors and then the top-$m=O(minPoints)$ nearest points to those random vectors. The resulting algorithm runs in $O(dk\cdot minPts)$ time and space to process each point, where dimension $d$ denotes the time for distance computations. If $k$ is constant, then with preprocessing, the total runtime is $O(d\cdot minPts + nD\log(D))$, which is subquadratic if $D = o(n)$. The space is $O(nk + D\cdot minPts)$, which is a good improvement.

To show their theoretical guarantees, they instead keep the top closest and furthest points ($R$ and $S$) to any random vector $r$. To see if a point is sufficiently close to $r$ to be in an $\epsilon$-ball, its distance is computed from all of $R$ and $S$. This boosts the probability that, given q is in the $\epsilon$-ball, from $1/2$ to $1-(1/2)^k$. In other words, they sample each edge in DBSCAN's resulting graph with probability $1-(1/2)^k$, which (by a known result) yields a graph with the same connected components for sufficiently large $k$. This argument holds in practice when they just use the top-$m$ points instead of these sets $R$ and $S$, though I was unsure of their reasoning.

Finally, due to sampling, this process may over-label points as noise. Afterwards, to combat this, they run a 1NN algorithm on perceived "noise" points to validate they are actually noise, using samples from all other points as training data. With sparse enough sampling, this added runtime is negligible.

They also test their theoretical findings with experiments. The main takeaway is that the new algorithm provides comparable runtime and accuracy on a single machine as k-means does on a supercomputer.

**Strengths:**

DBSCAN is a very popular and important clustering method with a wide impact. Any improvements to it, therefore, are important. The authors show nice, simple, and theoretically-backed methods to improve the efficiency of DBSCAN. While scalability has been studied before, they are the first to study it with theoretical guarantees on the quality of the scaled methods. The methods are non-trivial and interesting to see how they piece together to get these results. For the most part, the paper is nicely written.

**Weaknesses:**

There are a few parts in the paper that are dense and hard to understand. In addition, I'm unsure about the novelty of the work - which parts were actual new methodologies that they used? Many of their methods and lemmas seemed to cite past techniques. I put this in the Questions section so they may respond.

**Questions:**

1. "We observe that such memory constraint is the primary hurdle limiting the current scikit-learn implementation on million-point data
sets" -- This is presumably very dependent on hardware, input, etc, right? I feel like the way you've stated this is a bit too strongly. Could you say something more like "We observe that on our hardware, such memory..."?

2. In algorithm 2, why do you also need the furthest vectors?

3. In Theorem 1, you claim that you can assume t would be large because, otherwise, you should select different parameters anyways. What happens if t isn't large? You still need to be able to identify the parameters as bad selections. Does your algorithm allow identifying this if t isn't large, or do your assumptions fall apart?

4. I did not understand your selection of D - and I did not see the definition of D. Can you clarify: what is D, what it affects in Lemma 3, and why D=1,024 is adequate? Is D the dimension of the input data?

5. Can you clarify exactly what was a novel contribution and what was repurposed from prior works?

**Limitations:**

I do not see anything of concern.

---

> ### Author Rebuttal · Authors · 2024-08-05
>
> Thanks for your reviews. We will address the raised questions below.
>
> **Q1) "We observe that such memory constraint is the primary hurdle limiting the current scikit-learn implementation on million-point data sets" -- This is presumably very dependent on hardware, input, etc, right? I feel like the way you've stated this is a bit too strongly. Could you say something more like "We observe that on our hardware, such memory..."?**
>
> We agree and will fix it.
>
> For more specification, our machine has 128GB of RAM and tests on Pamap2 (n = 1.7M points) with size of 0.64 GB.
> We could not run spectral clustering, kernel k-means, DBSCAN, OPTICS provided by scikit-learn on Pamap2.
> For spectral clustering and kernel k-means, scikit-learn needs O(n^2) space to store pairwise distance matrix.
> We found from scikit-learn description: "This implementation has a worst case memory complexity of `O({n}^2)`, which can occur when the `eps` param is large and `minPts` is low, while the original DBSCAN only uses linear memory."
>
> Our implemented DBSCAN using linear memory works on Pamap2 but requires ~ 0.5 hours with multi-threading.
>
> **Q2) In algorithm 2, why do you also need the furthest vectors?**
>
> Given D random vectors $r_1, \ldots r_D$, if x is furthest to $r_1$, then $x$ must be closest to $-r_1$.
> We use the furthest vectors to partition the unit sphere with 2D vectors $r_1, \ldots, r_D$ and $-r_1, \ldots, -r_D$, and Lemma 1 still holds for top-k closest and furthest random vectors due to the symmetric Gaussian distribution.
>
> **Q3) In Theorem 1, you claim that you can assume t would be large because, otherwise, you should select different parameters anyways. What happens if t isn't large? You still need to be able to identify the parameters as bad selections. Does your algorithm allow identifying this if t isn't large, or do your assumptions fall apart?**
>
> We note that density-based clustering will link nearby core points together to form the cluster skeleton.
> Hence this assumption is needed to guarantee any density-based clustering.
> Consider the case where core points x, y where dist(x, y) = ε where B(x, ε) $\cap$ B(y, ε) = {x, y}.
> There is only **one** edge $xy$ connecting B(x, ε) and B(y, ε).
> In the worst case, if any approximate DBSCAN misses identifying $xy$ while approximating B(x, ε) and B(y, ε), it cannot recover DBSCAN's output.
> The assumption t = O(log(n)) makes the density-based cluster C not thin at anywhere, i.e. there are t paths $x p_i y$ that connect x, y together where $p_i$ are the core points in B(x, ε) $\cap$ B(y, ε).
>
> When minPts = 50 (note that t = log(8M) = 16), we expect many core points will have more than 16 core points within their ε-neighborhoods. This happens in practice since if x is in high-density regions, |B(x, ε)| >> minPts.
>
> If t is not large enough, there might be a thin region separating the density-based cluster.
> Then we should choose ε larger to increase the number of core points in ε-neighborhood, and hence increasing t.
> In practice, we use sOPTICS graphs to visualize the cluster structure and to select ε large enough to separate valleys.
> This will give good accuracy.
> Fig 1c, 1d shows that ε should be closer to 0.15 to separate 4 clusters, and Fig 1g, 1h shows that ε = 0.14 reaches the peak.
>
> A variant of assumption of t was used in [15] to guarantee the recovery of DBSCAN results of the sngDBSCAN.
> Besides, [15] also needs other 2 strong assumptions regarding the data distribution (See Assumption 1 in [15]) while we do not need them.
>
> **Q4) I did not understand your selection of D - and I did not see the definition of D. Can you clarify: what is D, what it affects in Lemma 3, and why D=1,024 is adequate? Is D the dimension of the input data?**
>
> D is the number of random vectors $r_i \sim N(0, 1)^d$ where d is input dimension.
> Indeed, Lemma 1 holds when D $\rightarrow$ ∞ , and in practice, we observe D = 1024 suffices for all 3 data sets.
> This setting was used in previous works [22, 23].
> On Lemma 3, we show that if $D = n^{1 / (k \alpha^2)}$ where $\alpha$ depends on the data distribution, and assume that such $D$ is large enough to ensure Lemma 1 holds, then we can find core points by maintaining just minPts points closest to each random vector.
> This saves the space and time complexity of sDBSCAN in practice.
>
> **Q5) Can you clarify exactly what was a novel contribution and what was repurposed from prior works?**
>
> We believe that the novel contribution is to enable density-based clustering (sDBSCAN) work on million-point data sets, and support many popular distance measures with a visual tool (sOPTICS) to guide the important parameter ε without ground truth.
> Both sDBSCAN and sOPTICS run fast and use small memory overheads compared to recent DBSCAN variants.
>
> We expect sDBSCAN will be as popular as k-means in clustering analysis due to its time and space complexity.
> sDBSCAN supports many distance measures, including L1, L2, $\chi^2$, JS, while k-means works on L2 distance, and kernel k-means and spectral clustering needs O(n^2) space to store the kernel matrix and the prior knowledge of k.
>
> Prior work scaling up DBSCAN limits on L2 [6, 12, 13, 14] or cannot achieve good accuracy [15].
> Especially, none of these works offer an efficient visual tool to select relevant parameter ε.
> We can see the execution time of sOPTICS + sDBSCAN is still much faster than sngDBSCAN [15] whereas an efficient heuristic to select relevant values of ε for sngDBSCAN seems difficult due to the natural of sampling.
>
> To prove theoretical guarantees for sDBSCAN, we use the recent result of Lemma 1 [22] in 2021 and the seminar result of Lemma 2 [27] in 1999.
> These theoretical foundations enable us to explain random landmark/pivots vectors and their neighborhoods are sufficient for density-based clustering.
> It works well in practice due to the small world phenomenon "Neighbors of neighbors tend to be neighbors".

---

> > ### Comment · Reviewer_C5J7 · 2024-08-12
> >
> > Thank you for your responses. Based off this and the other reviewers' comments, I will maintain my review score. I would more strongly defend this paper if I were more of an expert in this area, however I am still not confident about my review.

---

> > > ### Author Response · Authors · 2024-08-12
> > >
> > > Thanks for your comment. I hope you will engage in a discussion with ACs and other reviewers in the next stage to finalize the decision.

---

### Official Review · Reviewer_QDtd · 2024-07-12

**Soundness:** 3
**Presentation:** 3
**Contribution:** 3
**Rating:** 6
**Confidence:** 4

**Summary:**

Paper studies scalable algorithms for DBSCAN, a popular clustering method.
In DBSCAN, each point considers a radius of epsilon, and is called "core" if it has at least m points in the epsilon ball.
Then, each core point is connected to all other points (core or non-core) in its epsilon ball.

Finally, connected components are output as the clusters. Popular method used for clustering different types of datasets when k is not known, or the metric is not euclidean. However, slow to compute since identifying core points itself is cumbersome operation.

This paper tries to address the slowness (for euclidean vectors) by using random projections, a la LSH or locality sensitive hashing. Indeed, suppose all vectors are unit norm. Then suppose x is a database point and y is close to it, i.e., large inner product. We need to quickly identify if y is in the epsilon ball of x or not, and repeat this for all y, to determine if x is a core point or not. To this end, then suppose we draw many random directions in space, and we find that there is a direction r which has large inner product with both x and y. Then we can use r as a witness to conclude that y is likely to be in the ball of x.
To capture this, each point x will remember which are the top m random lines w.r.t inner product. And each line r will remember the top m' points w.r.t inner product. So when x will try to check if it is core or not, we dont scan with all points to see how many are in the epsilon ball. We simply scan the points which are in the top m' for the m random directions close to x, reducing the work needed to m*m' checks.
If we set m and m' suitably, we can prove that most of the real points are captured and this estimate works well.

The paper also shows theoretical guarantees under some assumptions on the connectivity of the clusters, that if the true connected components are sufficiently connected, then this algorithm will recover the connected components correctly.

The paper also gives emperical evals against SoTA implementations / approximations of DBSCAN and shows faster compute, and better Quality across different datasets.

**Strengths:**

DBSCAN seems like a popular tool, and making it more scalable is important problem.
Algo comes with theoretical backing, and is good empriically.
Paper is reasonably well written.

**Weaknesses:**

While authors try to compare with other works, not sure if the comparisons are clear enough as to why prior techniques dont do as well. In particular see the questions below.

**Questions:**

How do you extend this method to other metrics, non euclidean? I presume that one of the appeals of DBSCAN is the non-reliance of euclidean metric.
How does your method compare with LSH? So I will build some LSH and each point x will only look inside the bucket it lands into to check if it is core or not.
More generally I can use any SOTA approx. nearest neighbor method? Example HNSW or FAISS or SCANN? Will that not work?
Why dont you consider other real-world datasets generated by the Neural Networks, like OpenAI embeddings, etc? They represent a growing and fundamental workload right?

**Limitations:**

Would be good to include limitations section in paper.

---

> ### Author Rebuttal · Authors · 2024-08-05
>
> Thanks for your reviews. We will address the raised questions below.
>
> **Q1) How do you extend this method to other metrics, non euclidean? I presume that one of the appeals of DBSCAN is the non-reliance of euclidean metric.**
>
> sDBSCAN starts from cosine distance. To extend sDBSCAN on L1, L2, χ2, JS, we resort to kernel features f.
> We map x $\mapsto$ f(x) such that E[<f(x), f(y)>] = k(x, y) where k(., .) is a monotone function of other metrics.
> Extending sDBSCAN to non-metric distance is not trivial as the kernel matrix M is not positive-definite anymore.
> A possible solution is to use the feature embeddings into reproducing kernel **Krein** space [a, b].
> Since [b] can learn binary embeddings to preserve pair-wise non-metric distance, we can build sDBSCAN on such embeddings designed for non-metric distances.
>
> [a] Kernel Discriminant Analysis for Positive Definite and Indefinite Kernels - TPAMI 09
>
> [b] Non-Metric Locality-Sensitive Hashing - AAAI 10
>
>
> **Q2) How does your method compare with LSH? So I will build some LSH and each point x will only look inside the bucket it lands into to check if it is core or not.**
>
> While sDBSCAN and LSH [26] use random projections, they are indeed different.
> For sDBSCAN, for each random vector $r_i$, we find top-m' points closest to $r_i$. These points are used as the candidate set for any point x closest to $r_i$.
> The random vectors in sDBSCAN can be seen as random landmark/pivot points, and their neighborhoods (e.g. top-m' points) are useful to find the neighborhood of any point x closest to the pivot points.
> sDBSCAN indeed simulates the small world phenomenon "Neighbors of neighbors tend to be neighbors".
> For each point x, sDBSCAN finds top-m closet random vector to get m * m' candidates, hence the time complexity is O(dmm') and the extra space to store neighborhood is O(mm').
>
> For LSH, for each point x, we find the closest random vector.
> **All** points closest to the same random vector (i.e. bucket) will be the candidates.
> Hence, the size of the bucket corresponding to $r_i$ varies, e.g. large if $r_i$ points to high-density regions or small if $r_i$ points to low-density regions.
> We would need L = n^ρ hash tables to ensure good accuracy in finding core points where ρ is an LSH parameter.
> Building L = n^ρ hash tables requires O(n ^{1 + ρ}) space and O(dn ^{1 + ρ}) time.
> This cost is significant, especially compared to sDBSCAN's space and time complexity.
>
> To reduce the number of hash tables, practical multi-probing LSH schemes probe nearby buckets to get more candidates.
> Since we cannot control the collision probability of nearby buckets, there are likely many far-away points in the probing buckets.
> Compared to sDBSCAN, each point has to compute a fixed O(m*m') distances, and the preprocessing takes only around 10% of the total execution time (See Tab 4 in Appendix).
>
> Given a core point x, sDBSCAN tends to find the points closest to x, then approximates well the core distance (i.e. minPts-NN distance) of x.
> It is very useful since sOPTICS uses the core distance to compute the reachablity-distance, and to visualize the clustering structure (e.g. # valleys).
> Our sOPTICS shares similar execution time with sDBSCAN, and suggests the range of value of ε.
> We can see that sDBSCAN reaches the peak of accuracy given one value of ε in such range.
>
> Since LSH is designed for approximate ε-near neighbor search, it seems difficult to select potential closest points to x if the bucket of x is dense, hence it is challenging to have scalable OPTICS and DBSCAN variants with LSH.
>
> **Q3)  More generally I can use any SOTA approx. nearest neighbor method? Example HNSW or FAISS or SCANN? Will that not work?**
>
> Similar to LSH, the cost of constructing the index dominates the clustering time though there are still n queries to answer.
> HNSW builds an index in O(dn^2) time.
> We found that constructing HSNW data structure on 1.7M points (Pamap2) for small indexing time parameters (efConstruct = 128, m = 32) needs 7 seconds, which is similar to the running time of sDBSCAN.
> Building HNSW index on Mnist8m take more time than running sDBSCAN.
>
> Faiss and Scann need a Lloyd’s algorithm to learn the product quantization for the index.
> This learning phase requires O(dn * nlist) where nlist is the number of clusters.
> Compared to sDBSCAN with m = 10, m' = 50, then if nlist = 500, the learning phase has a similar cost of finding core points of sDBSCAN.
>
> We detailed the indexing time of Faiss, Scann, Hnsw on the response.pdf.
>
> sDBSCAN easily supports streaming data where you can insert or delete any point in the data set.
> For HNSW, Faiss and Scann, we would need to rebuild the data structure after deleting or inserting a sufficient number of points.
>
> **Q4) Why dont you consider other real-world datasets generated by the Neural Networks, like OpenAI embeddings, etc? They represent a growing and fundamental workload right?**
>
> Thank you for the suggestion.
> We indeed plan to run sDBSCAN on pre-trained image sets to evaluate its utility in practical applications.
> However, our concern is whether we take into account the time of computing embeddings into the preprocessing time.
> For example, we currently test Mnist/Mnist8m data set whose dimensions represent pixel values.
> If no, then we consider the pre-trained data set as a new data set.
> If yes, then extracting the OpenAI embeddings will dominate the running time of clustering, especially with large models.

---

### Author Rebuttal · Authors · 2024-08-07

**1) Selecting $\epsilon$ for sDBSCAN by sOPTICS without ground truth**

We would like to use global rebuttal to explain further sOPTICS  (details in Section A1), one of our contributions, to select the parameter $\epsilon$ of sDBSCAN.
Without sOPTICS, selecting a good value for $\epsilon$ will be as inefficient as running DBSCAN exactly.

On OPTICS graphs, the x-axis is the point ID and the y-axis presents its corresponding reachability distance ($reachDist$). The points are ordered based on the cluster structure, e.g. if $x$ and $y$ are close, and tend to be in the same cluster, then $x$ and $y$ will be ordered nearby on x-axis.

OPTICS assigns a distance called $reachDist$ for each point.
The $reachDist$ of core points are their $minPts$-NN distances.
Core points in high-density regions will have smaller $reachDist$ while core points in low-density regions will have larger $reachDist$.
The $reachDist$ of a non-core point $q$ tends to be $dist(q, x)$ where $x$ is the closest core point to $q$.

In brief, OPTICS starts the process from a random point, and looks up the core points nearby the processed points so far.
Core points with the smallest $reachDist$ (stored in a priority queue) tends to be processed first, and output to a clustering order.
Hence, the points tend to be grouped with its neighborhood.
A sharp decrease of $reachDist$ indicates that we are processing points in the denser regions, and a slightly increase of $reachDist$ indicates that we are processing points in sparser regions.
This creates the valley, which is used as the cluster identification.

Similar to OPTICS, the number of valleys in the sOPTICS dendrograms reflect the number of clusters, and points downwards the valley floor are on denser regions while points upwards the valley head are on sparser regions.
By selecting $\epsilon$ to separate these valleys, sDBSCAN will achieve the peak of accuracy.

sOPTICS also needs ($\epsilon'$, minPts) to run and such $\epsilon'$ is usually larger than $\epsilon$ of sDBSCAN.
This ensures $reachDist$ of a point is its minPts-NN distance, which reflects its local density.
This links to the assumption of $t$ used in Theorem 1.
A wrong choice of $\epsilon$ for sDBSCAN will break the cluster into many smaller clusters (due to small $t$ at some place of the cluster).
We should increase $\epsilon$ to ensure $t$ large enough.
The $\epsilon$ nearby the valley head tends to give the peak of clustering accuracy (see Fig 1c-d for sOPTICS and Fig 1g-h for sDBSCAN's accuracy) since it strengthens the clustering structure on dense regions (i.e. most of points in $B(x, \epsilon)$ are core points).

sngDBSCAN [15] uses a somewhat stronger variant of the assumption of $t$ to ensure the recovery of DBSCAN's output.
However, none of DBSCAN variants offers an efficient tool to select a relevant value of $\epsilon$.

**2) Limitations in theory**

Our theoretical guarantees on recovering DBSCAN's output need $D \rightarrow \infty$ where $D$ is the number of random vectors, which is our limitation.
However, sDBSCAN works well in practice with $D = 1024$ even for Mnist8m ($n$ = 8.1M, $d$ = 784).
This might be due to the small world phenomenon "Neighbors of neighbors tend to be neighbors".

We use random vectors $r_i$ as pivot points and consider the top-$m$ points closest to $r_i$ as $r_i$'s neighbors.
If $x$ and $y$ are closest to the same random vectors $r_i$, they tend to be close to points in $r_i$'s neighbors.
That will increase the clustering connection when core points $x$ and $y$ share the same neighbors.

**3) Practicality**

For practical setting $D = 1024$, $m = minPts$ and constant $k$, sDBSCAN and sOPTICS (support various metric distances) use small memory overheads (i.e. $O(n * minPts)$) to store $n$'s neighborhods and their main computation cost is distance computations (i.e. $O(dn * minPts)$).

sDBSCAN with recommended parameter provided by sOPTICS is favoured compared to other clustering algorithms, including k-means variant, kernel k-means, sampling-based DBSCAN on million-point high-dimensional data sets, demonstrated by our empirical results in the paper and appendix.

**Response.pdf**

We add a figure to measure the index construction time of approximate nearest neighbor search on SOTA industry libraries, including Faiss, Scann, Hnsw.

We add AMI and NMI measures for the clustering accuracy.

---

### Decision · Program_Chairs · 2024-09-25

**Decision:**

Accept (poster)

**Comment:**

The paper introduces sDBSCAN, a scalable density-based clustering algorithm in high dimensions with cosine distance. During the review period the following strength and weaknesses have been highlighted.

Strengths:
- the papers improves the scalability of an important algorithm
- the algorithm comes with theoretical backing, and is good empirically
- The methods are non-trivial and interesting

Weaknesses:

- There are a few parts in the paper that are dense and hard to understand
- Limited novelty, the core idea of approximating-neighborhood is borrowed from the previous work [22]

Overall, the paper is interesting and it may be good to accept as a poster.

The committee strongly suggest the authors to implement the discussed improvements regarding the formulation of the theoretical results